# FL118 Is a Potent Therapeutic Agent against Chronic Myeloid Leukemia Resistant to BCR-ABL Inhibitors through Targeting RNA Helicase DDX5

**DOI:** 10.3390/ijms25073693

**Published:** 2024-03-26

**Authors:** Kengo Takeda, Satoshi Ohta, Miu Nagao, Erika Kobayashi, Kenji Tago, Megumi Funakoshi-Tago

**Affiliations:** 1Division of Hygienic Chemistry, Faculty of Pharmacy, Keio University, 1-5-30 Shibakoen, Minato-ku, Tokyo 105-8512, Japan; taken50502@keio.jp (K.T.); nagaomiu@keio.jp (M.N.); erica.kobayashi@keio.jp (E.K.); 2Division of Structural Biochemistry, Department of Biochemistry, School of Medicine, Jichi Medical University, 3311-1 Yakushiji, Shimotsuke-shi 329-0498, Tochigi, Japan; satoshi.ohta@jichi.ac.jp; 3Department of Laboratory Sciences, Gunma University Graduate School of Health Sciences, 3-39-22 Showa-Machi, Maebashi 371-8514, Gunma, Japan; ktago@gunma-u.ac.jp

**Keywords:** chronic myeloid leukemia (CML), BCR-ABL, DDX5, FL118, apoptosis

## Abstract

Chronic myeloid leukemia (CML) is induced by the expression of the fused tyrosine kinase BCR-ABL, which is caused by a chromosomal translocation. BCR-ABL inhibitors have been used to treat CML; however, the acquisition of resistance by CML cells during treatment is a serious issue. We herein demonstrated that BCR-ABL induced the expression of the RNA helicase DDX5 in K562 cells derived from CML patients in a manner that was dependent on its kinase activity, which resulted in cell proliferation and survival. The knockout of DDX5 decreased the expression of BIRC5 (survivin) and activated caspase 3, leading to apoptosis in K562 cells. Similar results were obtained in cells treated with FL118, an inhibitor of DDX5 and a derivative compound of camptothecin (CPT). Furthermore, FL118 potently induced apoptosis not only in Ba/F3 cells expressing BCR-ABL, but also in those expressing the BCR-ABL T315I mutant, which is resistant to BCR-ABL inhibitors. Collectively, these results revealed that DDX5 is a critical therapeutic target in CML and that FL118 is an effective candidate compound for the treatment of BCR-ABL inhibitor-resistant CML.

## 1. Introduction

Chronic myeloid leukemia (CML) is a type of leukemia that is characterized by the presence of the Philadelphia chromosome, resulting from the chromosomal translocation t (9; 22) (q34; q11). The Philadelphia chromosome encodes *BCR-ABL*, a fusion gene consisting of a partial region of the breakpoint cluster region (*Bcr*) and a partial region of the Abelson (*Abl*) tyrosine kinase corresponding to its kinase domain [1,2,3]. The expressed fusion protein BCR-ABL forms oligomers, and thereby undergoes autophosphorylation and functions as a constitutively active tyrosine kinase [4]. BCR-ABL has been identified as a causative gene product of CML through its stimulation of cellular proliferative signals, including the transcription factor signal transducer and activator of transcription 5 (STAT5) and the mitogen-activated protein kinase/extracellular signal-regulated kinase kinase extracellular signal-regulated kinase (MEK-ERK) pathway [5,6,7]. BCR-ABL has been shown to directly phosphorylate STAT5 on tyrosine 694 (Y694) and augment its transcriptional activity [8,9]. The phosphorylation of BCR-ABL at tyrosine residue 117 (Y177) activates the small GTP protein Ras, which is mediated by the recruitment of the activator protein complex containing GRB2 and SOS, and subsequently activates the MEK-ERK pathway [10,11]. BIRC5, also known as survivin, a member of the inhibitor of apoptosis (IAP) family, functions to inhibit caspase activation, thereby leading to the negative regulation of apoptosis [12]. *BIRC5* is one of the target genes of STAT5, and its expression is also induced by the Ras signaling pathway [13,14]. In cells transformed with BCR-ABL, antiapoptotic signals are activated through the up-regulated expression of BIRC5 [15]. Therefore, the application of inhibitors targeting BCR-ABL is a major therapeutic strategy in the treatment of CML.

Imatinib, a competitive inhibitor that binds to the ATP-binding site of BCR-ABL, is a first-generation BCR-ABL inhibitor that is highly effective against CML [16,17]. However, CML patients often acquire resistance to imatinib due to the emergence of various mutations in the BCR-ABL gene during treatment, particularly in the region encoding the kinase domain of ABL [18]. Therefore, the second-generation BCR-ABL inhibitors, dasatinib and nilotinib were developed to overcome this resistance to imatinib, making it possible to treat CML patients with many *BCR-ABL* gene mutations [19,20]. However, among *BCR-ABL* gene mutations, the T315I mutation, called the gatekeeper mutation, in which threonine at position 315 is replaced with isoleucine, exhibits resistance to not only imatinib but also dasatinib and nilotinib because its introduction into BCR-ABL inhibits its binding with these BCR-ABL inhibitors [21,22]. Ponatinib has been developed as a third-generation BCR-ABL inhibitor that suppresses BCR-ABL mutants with the T315I mutation. However, ponatinib frequently causes cardiovascular disorders, and the Food and Drug Administration ordered a moratorium on its usage for the treatment with CML in 2013 [23,24]. Therefore, the development of novel CML treatments that are effective against BCR-ABL with the T315I mutation and have fewer side effects is still required.

DDX5 is a member of the RNA helicase family that is characterized by the Asp-Glu-Ala-Asp motif (DEAD-box) and is involved in many steps of the gene expression process, including RNA processing, DNA replication, ribosome biosynthesis, microRNA maturation, and transcriptional regulation [25,26,27,28,29,30]. The expression of DDX5 is up-regulated in various types of cancers, and its functional involvement in processes for the malignant transformation of tumors, such as invasion and metastasis, has been demonstrated [31]. Iyer and colleagues found that combination of DDX5 and polo-like kinase-1 (PLK1) positive cases in p53 mutant breast cancer exhibited poor prognosis [32]. Dai et al. reported that high expression of DDX5 and fructose-bisphosphate aldolase A (ALDOA) are associated with poor prognosis in colorectal cancer (CRC) patients [33]. In addition, it was reported that concomitant long non-coding RNA (lncRNA) *NEAT1* and DDX5 protein levels negatively correlated with the survival of CRC patients [34]. We also observed the correlated high expression of DDX5 and a transcription factor c-Myc in CRC patients and found the presence of their oncogenic positive feedback loop [35]. According to these reports, despite the lack of detailed molecular mechanisms, the importance of DDX5 in cancer development and malignant transformation has been demonstrated by many researchers. Furthermore, numerous studies reported the contribution of DDX5 to tumorigenesis-related gene expression through its function as a coactivator for several transcription factors, including β-catenin/transcription factor 4 (TCF4), NF-κB, c-Myc, and Fra-1 [35,36,37,38,39]. The Wnt-β-catenin/TCF4 signaling pathway was found to be abnormally activated in colon and breast cancers and associated with a poor prognosis [40], suggesting the potential of DDX5 as a valuable diagnostic and prognostic marker and potential therapeutic target for many tumors.

In 2022, Ling and colleagues reported FL118, a derivative of camptothecin (CPT) and inhibitor of topoisomerase I (Topo I), as a novel DDX5 inhibitor [41]. FL118 was originally discovered by screening a compound library using the *BIRC5* reporter system and was shown to exhibit excellent antitumor activity in human tumor xenograft models [42]. It was newly revealed that FL118 binds with DDX5, thereby suppressing its phosphorylation and inducing its degradation by the ubiquitin–proteasome system [41].

In the present study, we examined the role of DDX5 in K562 cells derived from BCR-ABL-positive CML patients through its genetic deficiency and usage of the DDX5 inhibitor FL118. We also investigated whether the deletion or inhibition of DDX5 was effective in Ba/F3 cells transformed with the BCR-ABL T315I mutant, which is resistant to BCR-ABL inhibitors, and verified the potential of CML treatment targeting DDX5.

## 2. Results

### 2.1. The Expression of DDX5 Was Reduced by Treatment with Imatinib in K562 Cells

To investigate whether DDX5 is downstream of BCR-ABL in CML cells, we examined the effects of the BCR-ABL inhibitor imatinib on the expression of DDX5 in K562 cells. Immunoblotting confirmed that the tyrosine phosphorylation of BCR-ABL and its substrate, STAT5, was prevented by the treatment with imatinib (Figure 1A). The expression of DDX5 mRNA was significantly reduced by treatment with imatinib in K562 cells, suggesting that the regulation of DDX5 by BCR-ABL is partially due to the transcriptional up-regulation (Figure 1B). The expression of BIRC5 and *BIRC5* mRNA was significantly reduced by the treatment with imatinib (Figure 1A,C). In addition, the treatment with imatinib induced the cleavage of caspase-3, suggesting that imatinib induced apoptosis in K562 cells (Figure 1A). The treatment with imatinib significantly attenuated the proliferation rate, and the IC_50_ of imatinib was 0.42 µM (Figure 1D). Furthermore, the treatment with imatinib significantly reduced the viability of K562 cells (Figure 1E). Under treatment conditions with imatinib, the expression of DDX5 was significantly reduced (Figure 1F). There was no significant difference in the suppressive effects of 1 µM and 2 µM imatinib on the proliferation ability and viability of K562 cells and on the expression of DDX5 (Figure 1B,D–F). On the other hand, a significant difference was observed between the treatment with 1 µM and 2 µM imatinib in suppressing the BIRC5 expression and *BIRC5* mRNA (Figure 1A,C). These results suggest that the kinase activity of BCR-ABL is essential for the expression of DDX5 in K562 cells.

### 2.2. DDX5 Was Essetial for the Proliferation and Survival of K562 Cells

To examine the role of DDX5 in K562 cells, we generated K562 cells lacking DDX5 (DDX5-KO) using genome editing methods (Figure 2A). After control K562 cells and DDX5-KO cells were incubated without IL-3 for 24 h, the protein and mRNA expression of BIRC5 was significantly reduced in DDX5-KO cells (Figure 2A,B). The activation of caspase-3 was observed in DDX5-KO cells (Figure 2A). Furthermore, the proliferative ability and viability of DDX5-KO cells were significantly lower than those of control K562 cells (Figure 2B,C). These results suggest that DDX5 is an essential molecule for the proliferation and survival of K562 cells.

### 2.3. The CPT Derivative FL118 Suppressed the Expression of DDX5 in K562 Cells

FL118 has a structure in which a methylenedioxy group is introduced into CPT (Figure 3A). We used recombinant Topo I and supercoiled DNA to measure the inhibitory effects of CPT and FL118 on Topo I. When supercoiled DNA was mixed with Topo I, it was converted to relaxed DNA. Furthermore, when CPT or FL118 was added, the conversion of supercoiled DNA to relaxed DNA by Topo I was suppressed. Furthermore, no marked differences were observed in the inhibition of Topo I between CPT and FL118 (Figure 3B). When K562 cells were treated with CPT or FL118 and their effects on DDX5 expression were examined by immunoblotting, only FL118 exerted suppressive effects (Figure 3C). Furthermore, the FL118-induced suppression of DDX5 expression was significantly inhibited by the proteasome inhibitor MG132 in K562 cells (Figure 3D), suggesting that FL118 induced the proteasomal degradation of DDX5. Therefore, FL118 specifically possessed the ability to inhibit the expression of DDX5 in addition to Topo I.

### 2.4. FL118 Induced Apoptosis More Strongly Than CPT in K562 Cells

We examined the ability of CPT and FL118 to induce cell death in K562 cells. The treatment with CPT reduced both the protein and mRNA expression of BIRC5 in a dose-dependent manner and induced the activation of caspase-3 in K562 cells. The treatment with FL118 decreased both the protein and mRNA expression of BIRC5 and induced the activation of caspase-3 more strongly than the treatment with CPT (Figure 4A,B). Furthermore, the treatment with FL118 suppressed the proliferative ability of K562 cells at lower concentrations than the treatment with CPT, with IC_50_ of 229.4 and 51.9 nM for CPT and FL118, respectively. The treatment with FL118 was also more cytotoxic against K562 cells than the treatment with CPT (Figure 5A,B). These results revealed that FL118 effectively induced apoptosis by down-regulating the expression of DDX5 in K562 cells.

### 2.5. The BCR-ABL T315I Mutant, Which Is Resistant to Imatinib, Induced the Expression of DDX5, Similar to BCR-ABL

The BCR-ABL mutant harboring the T315I mutation shown in Figure 6A was previously reported to be resistant to BCR-ABL inhibitors [21,22]. We generated Ba/F3 cells expressing BCR-ABL and the BCR-ABL T315I mutant by retroviral infection and examined the expression of DDX5 in these cells. The enforced expression of BCR-ABL and BCR-ABL T315I mutant significantly increased the expression of DDX5 in Ba/F3 cells (Figure 6B). Then, we examined the sensitivities of these cells to imatinib. In Ba/F3 cells expressing BCR-ABL, the treatment with imatinib suppressed the phosphorylation of BCR-ABL and STAT5. Furthermore, in Ba/F3 cells expressing BCR-ABL, the treatment with imatinib reduced the protein and mRNA expression of BIRC5 and induced the cleavage of caspase-3. In contrast, in Ba/F3 cells expressing the BCR-ABL T315I mutant, the treatment with imatinib did not affect the phosphorylation of BCR-ABL or STAT5 or the protein or mRNA expression of BIRC5 and also did not induce the activation of caspase-3 (Figure 6C). The treatment with imatinib reduced the expression of DDX5 in Ba/F3 cells expressing BCR-ABL but did not affect the expression of DDX5 in Ba/F3 cells expressing the BCR-ABL T315I mutant (Figure 6D). The treatment with imatinib inhibited the proliferative ability of Ba/F3 cells expressing BCR-ABL in a concentration-dependent manner, with an IC_50_ of 0.62 μM. On the other hand, imatinib did not affect the proliferative ability of Ba/F3 cells expressing the BCR-ABL T315I mutant (Figure 6E). Furthermore, imatinib significantly decreased the viability of Ba/F3 cells expressing BCR-ABL but did not affect the viability of Ba/F3 cells expressing the BCR-ABL T315I mutant (Figure 6F).

### 2.6. The Knockout of DDX5 Reduced the Proliferation of and Induced Cell Death in Ba/F3 Cells Expressing the BCR-ABL T315I Mutant

We investigated the roles of DDX5 in Ba/F3 cells expressing BCR-ABL or the BCR-ABL T315I mutant by its deletion using genome editing methods. The expression of BIRC5 and Birc5 mRNA were significantly decreased by the knockout of DDX5 in both cells (Figure 7A). Under these conditions, the activation of caspase-3 was not detected. However, the proliferative ability and viability of both cells were significantly attenuated by the knockout of DDX5 (Figure 7B,C). These results suggest that the expression of DDX5 was induced not only by BCR-ABL, but also by the BCR-ABL T315I mutant, and that DDX5 was involved in cell proliferation and survival, even in CML cells resistant to BCR-ABL inhibitors.

### 2.7. FL118 Induced Apoptosis Not Only in Ba/F3 Cells Expressing BCR-ABL, but Also in Ba/F3 Cells Expressing the BCR-ABL T315I Mutant

We examined the effects of CPT and FL118 on Ba/F3 cells expressing BCR-ABL or the BCR-ABL T315I mutant. In both cells, the treatment with CPT did not affect the expression of DDX5, whereas that with FL118 suppressed its expression (Figure 8A). Furthermore, while the treatment with FL118 significantly suppressed both BIRC5 expression and Birc5 mRNA expression and strongly induced the activation of caspase-3, the treatment with CPT failed to exert similar effects (Figure 8B). The treatment with CPT suppressed the proliferative ability and reduced the viability of Ba/F3 cells expressing BCR-ABL and those expressing the T315I mutant to a similar extent. The treatment with FL118 suppressed the proliferative ability of these cells more strongly than the treatment with CPT. The IC_50_ of CPT and FL118 against Ba/F3 cells expressing BCR-ABL were 122.0 and 5.97 nM, respectively. The IC_50_ of CPT and FL118 against Ba/F3 cells expressing the BCR-ABL T315I mutant were 98.3 and 6.23 nM, respectively (Figure 9A). Furthermore, the treatment with FL118 induced cell death in both Ba/F3 cells expressing BCR-ABL and those expressing the BCR-ABL T315I mutant more strongly than the treatment with CPT (Figure 9B). These results indicate that FL118, which suppressed the expression of DDX5, exhibited high antitumor activity, even against CML resistant to BCR-ABL inhibitors. When considering the mechanism of action of FL118, it is necessary to discuss the inhibitory effect of FL118 on Topo I, as the treatment with CPT has been reported to cause apoptosis through activation of caspase-3 [43]. Therefore, it is thought that FL118 exhibits cytotoxicity against CML cells by not only suppressing DDX5 expression but also inhibiting Topo I (Figure 10).

## 3. Discussion

The development of effective therapeutic agents to treat cases of CML exhibiting resistance to existing BCR-ABL inhibitors, such as imatinib, is an important issue that cannot be overlooked. The development of imatinib as a therapeutic agent for CML increased the survival rate of these patients [16,17]. However, cases of CML exhibiting resistance to imatinib appeared during treatment, particularly at the blast crisis [44,45]. When entering the blast crisis, several additional genetic mutations or modifications are induced in tumor suppressor genes, such as p53, RB, and related p16^INK4a^ and Arf. The protein expression of RB was lacking in the blast crisis of CML [46,47]. RB binds and suppresses the functions of transcription factors in the E2F family, which induces cell cycle entry from the G_1_ to S phase. Other studies highlighted the presence of mutations in the TP53 gene. TP53 forms a homotetramer as an active transcription factor, and induces the expression of its target genes, such as p21^CIP1^ and PUMA [48,49,50]. When entering into the blast crisis of CML, the TP53 gene is reportedly mutated, resulting in the production of a dysfunctional TP53 protein acting as a dominant negative mutant [51]. In addition, p16^INK4a^ and Arf harbor critical roles for the function and activation of RB and TP53; however, the expression of p16^INK4a^ and Arf was shown to be silenced by the DNA methylation of their promoter sequences [52].

A number of molecular mechanisms are responsible for the acquisition of resistance to BCR-ABL inhibitors. Hochhaus and colleagues investigated 66 imatinib-resistant CML patients [53]. They evaluated these patients for the mRNA expression of BCR-ABL, the genomic amplification of BCR-ABL, clonal karyotypic evolution, and mutations in the BCR-ABL tyrosine kinase domain. Four main findings were obtained as follows: (1) 12% of imatinib-resistant CML patients showed >10-fold increases in BCR-ABL levels; (2) 6% showed the genomic amplification of BCR-ABL by fluorescence in situ hybridization; (3) additional chromosomal aberrations were observed in 53% of patients; (4) point mutations in the ABL tyrosine kinase domain were detected in 35% of imatinib-resistant CML patients. The T315I mutation was previously reported to be ineffective against imatinib and also induced resistance against second-generation nilotinib and dasatinib [21,22].

The novel BCR-ABL inhibitor, asciminib, was recently developed and has a different mechanism of action to existing BCR-ABL inhibitors. Asciminib binds to the myristoyl pocket of ABL and allosterically inhibits kinase activity, and is effective against the BCR-ABL T315I mutant [54]. Although advances have been achieved in the treatment of CML patients with resistant mutations in the BCR-ABL kinase domain, resistance has not yet been completely overcome because additional resistant mutations against this novel Bcr-ABL inhibitor may emerge. Therefore, to identify novel therapeutic targets, further analyses of downstream signals of BCR-ABL are important.

In the present study, we confirmed the necessity of DDX5 in the survival and proliferation of CML cells induced by BCR-ABL, and showed for the first time that therapy targeting DDX5 was effective for the treatment of CML that acquired drug resistance (Figure 8). However, the mechanisms by which DDX5 contributes to BCR-ABL-induced oncogenic signaling have not yet been elucidated. We previously found that DDX5 played an essential role in the signaling pathway of the tyrosine kinase Janus kinase 2 (JAK2) V617F mutant, which causes myeloproliferative neoplasms (MPN). Although the underlying molecular mechanisms have yet to be clarified, the activation of STAT5, a downstream signaling molecule of the JAK2V617F mutant, was required for the stabilization of the DDX5 protein in MPN cells [55]. Since BCR-ABL also activates STAT5 [56,57], these findings suggest that the expression of DDX5 may be induced in CML cells via a similar mechanism. We also reported that DDX5 interacted with several transcription factors, including the glucocorticoid receptor (GR) and c-Myc, and induced their transcriptional activation [35,58]. Furthermore, we confirmed that the RNA helicase activity of DDX5 was required for its function as a coactivator for GR. On the other hand, DHX15, another RNA helicase that possesses structural similarities with DDX5, interacted with MYC and contributed to the stability of the MYC protein at the post-translational level; however, its RNA helicase activity was dispensable [59]. Therefore, the RNA helicase activity of DDX5 may not be required for some of its functions.

We first investigated the effects of imatinib on proliferation, survival, and DDX5 expression in K562 cells. With reference to previous reports [60,61], we chose two concentrations (1 and 2 µM) of imatinib to test their effects. However, there was no significant difference in the suppressive effects of 1 µM and 2 µM imatinib on them (Figure 1). It has been reported that K562 cells are a mixture of cell populations exhibiting different sensitivity to imatinib and that an imatinib-resistant subpopulation is adherent and has up-regulated expression of BCR-ABL [62]. This is thought to be the reason why no clear concentration dependence of the effect of imatinib was observed in Figure 1.

In this study, we examined the antitumor activity of FL118 and CPT against K562 cells and Ba/F3 cells expressing BCR-ABL or its T315I mutant (Figure 4 and Figure 8). Through the experiments, both common and different effects of FL118 and CPT were found, and some of them had discrepant results that are difficult to explain the circumstances. The treatment with CPT suppressed the expression of *BIRC5* mRNA and BIRC5 protein in K562 cells, although it was weaker than the FL118 treatment (Figure 4). On the other hand, in Ba/F3 cells expressing BCR-ABL or its T315I mutant, the treatment with CPT did not suppress the expression of *Birc5* mRNA or BIRC5 protein, and the treatment with FL118 only significantly suppressed these expressions (Figure 8). It has been reported that CPT activates the tumor suppressor p53 and that the inactivation of p53 increases the cytotoxicity of CPT in cancer cells [63,64]. Although p53 in Ba/F3 cells is intact, p53 in K562 cells has a genetic mutation and is inactivated [65]. Therefore, it is possible that CPT had a strong suppressive effect on BIRC5 expression in K562 cells. However, on the other hand, it was previously reported that p53 transcriptionally represses BIRC5 expression by interfering with E2F1 on the *BIRC5* promoter or by the recruitment of a corepressor factor Sin-3 and histone deacetylases HDAC on the *BIRC5* promoter [66,67]. Now, we have no suitable explanation for this discrepant result.

The mechanisms by which the activation of DDX5 is regulated have not yet been elucidated. Yang and colleagues reported the tyrosine phosphorylation of DDX5 in K562 cells as well as various cancer cells, including lung, colorectal, liver, breast, and cervical cancers, but not in untransformed cells [68]. Furthermore, under a platelet-derived growth factor (PDGF) stimulation, DDX5 was phosphorylated at Y593 via c-Abl and then promoted the nuclear translocation of β-catenin through a Wnt-independent pathway, leading to epithelial–mesenchymal transition [69]. Although we attempted to observe the tyrosine phosphorylation of DDX5 at Y593 in K562 cells by immunoblotting, we failed to see it. It is unclear whether its phosphorylation at Y593 causes very low level or other tyrosine phosphorylation of DDX5 functions to regulate the activity of DDX5. On the other hand, a treatment with oxaliplatin, an anticancer drug classified as a platinum preparation, phosphorylates DDX5 at T564 and/or T446 via p38, thereby inducing apoptosis in colon cancer cells [58]. These findings suggest that the function of DDX5 is both positively and negatively regulated by the phosphorylation of tyrosine and threonine residues. In the current study, we could conclude that FL118 could be utilized for the treatment of CML resistant to BCR-ABL inhibitor. The functional roles of DDX5 phosphorylation should be clarified in future projects.

In the current study, we analyzed the expression of BIRC5 as a marker to evaluate the inhibitory effect of FL118 on DDX5. Originally, the functional relationship between DDX5 and BIRC5 was reported by Ling and colleagues [41]. They found that BIRC5 expression was attenuated when the expression of DDX5 was silenced. As described above, it has been well established that DDX5 interacts with various transcription factors and regulates their activity, therefore there is no doubt for considering the involvement of DDX5 in the transcriptional regulation of *BIRC5* promoter. This issue will be advanced in the future as the molecular mechanisms of *BIRC5* promoter regulation are elucidated.

Our observations failed to clarify whether FL118 could affect the RNA helicase activity of DDX5, but the effect of FL118 on the RNA helicase activity should be clarified in near future analysis. To clarify, we need to purify the active recombinant protein of DDX5 and perform an in vitro RNA helicase assay. In the current study, we showed that the expression of DDX5 protein decreased when treated with FL118. Furthermore, the knockout of DDX5 showed comparable inhibitory effects on cell proliferation and BIRC5 expression in both K562 cells and the Ba/F3 cells expressing BCR-ABL, as did the treatment with FL118. Therefore, it is thought that the effect of FL118 was more likely to be due to the degradation of DDX5 than the inhibition of the helicase activity of DDX5. We also have previously expressed wild-type DDX5 and the DDX5 mutant (K144N) lacking RNA helicase activity in adipocyte precursor 3T3-L1 cells and observed that their expression levels were almost the same [58]. These results suggested that there is no correlation between the RNA helicase activity of DDX5 and the regulation of its expression. According to a previous report by Ling and colleagues, DDX5 was identified as an FL118 binding protein, and FL118 behaves as a molecular glue degrader against DDX5 [41]. This report suggests to us that FL118-caused down-regulation of DDX5 was most unlikely due to deceleration of protein synthesis of DDX5. In fact, we observed that the FL118-induced decrease in DDX5 expression was significantly restored by MG132 treatment in K562 cells (Figure 3D). Therefore, FL118 may promote proteasomal degradation of DDX5 in K562 cells. The detailed mechanism of how DDX5 is down-regulated by the treatment with FL118 will be a problem that should be clarified in the next project.

In the absence of IL-3, the expression of DDX5 was slightly observed in Ba/F3 cells, but the enforced expression of BCR-ABL significantly induced DDX5 expression (Figure 6B). Considering this phenomenon and the results obtained with DDX knockout and treatment with FL118, we proposed a molecular mechanism by which enhanced expression of DDX5 positively regulates proliferation and survival of BCR-ABL-positive CML cells (Figure 10). However, we only analyzed the function of her DDX5 in K562 cells and artificial BCR-ABL expressing Ba/F3 cells in this study. In future studies, we need to demonstrate the importance of DDX5 in CML pathogenesis using other BCR-ABL-positive leukemic cell lines such as KYO1, LAMA84, EM2, EM3, BV173, AR230, KU812, and KCL22. Furthermore, it is necessary to examine the effect of FL118 on CML progression using CML model mice and verify its therapeutic possibility.

Accumulated evidence emphasizes the importance of investigating the molecular mechanisms underlying the activation of DDX5, including the phosphorylation of DDX5. DDX5 has been reported to be involved in RNA metabolisms and to function as a coactivator for certain transcription factors [25,26,27,28,29,30] and is thought to be deeply related to gene expression regulation downstream of BCR-ABL in K562 cells. In the future, it will be necessary to analyze how the knockout of DDX5 in K562 cells affects gene expression by RNA sequence analysis and to understand the DDX5-mediated carcinogenesis mechanism activated by BCR-ABL. Further research on therapeutic combinations of inhibitors targeting downstream signals of BCR-ABL along with BCR-ABL inhibitors will be an important therapeutic strategy for CML.

## 4. Materials and Methods

### 4.1. Reagents and Antibodies

Recombinant murine interleukin-3 (IL-3) and puromycin were purchased from PEPROTECH (Rocky Hill, NJ, USA) and InVivoGen (San Diego, CA, USA), respectively. Imatinib, FL118, and CPT were purchased from Cayman Chemical (Ann Arbor, MI, USA), AbMole Bioscience (Houston, TX, USA), and Adooq Bioscience LLC (Irvine, CA, USA), respectively. MG132 was purchased from Nacalai Tesque (Kyoto, Japan). An antiDDX5 antibody (05-850) was purchased from Merck Millipore (Darmstadt, Germany). Antibodies against phospho-Abl (#2861), c-Abl (#2862), phospho-STAT5 (#9351), and cleaved caspase-3 (#9661) were obtained from Cell Signaling Technologies (Danvers, MA, USA). Antibodies against STAT5 (sc-74442), BIRC5/survivin (sc-17779), and β-actin (sc-47778) and an anticaspase-3 (p17) antibody (sc-271028) to detect human cleaved caspase-3 were purchased from Santa Cruz Biotechnology (Santa Cruz, CA, USA). Horseradish peroxidase (HRP)-conjugated secondary antibodies (21860-61, 21858-11) and other reagents were purchased from Nacalai Tesque (Kyoto, Japan).

### 4.2. Plasmids

pSG5-P190 was a gift from Nora Heisterkamp (Addgene plasmid # 31285; http://n2t.net/addgene:31285; accessed on 1 January 2018. RRID:Addgene_31285). The cDNA encoding p190 BCR-ABL was subcloned into the MSCV-IRES-GFP retroviral vector. The substitution of the amino acid residue T315I in BCR-ABL was performed using a site-directed mutagenesis kit (Stratagene, CA, USA) as previously described [59]. lentiCRISPR v2 was a gift from Feng Zhang (Addgene plasmid #52961; http://n2t.net/addgene:52961; accessed on 1 April 2023. RRID:Addgene_52961). The target sequences of human DDX5 (5′-CCATGTCGGGTTATTCGAGTGAC-3′) and mouse DDX5 (5′-GAGACCGCGGCCGGGATCGAGGG-3′) were designed using CRISPRdirect (http://crispr.dbcls.jp; accessed on 1 August 2023). Annealed oligos were cloned into lentiCRISPR v2 and clones were confirmed by Sanger DNA sequencing using a primer (5′-GGACTATCATATGCTTACCG-3′) of the U6 promoter. The helper vectors, pGP Vector and pE-eco (Takara Bio Inc., Shiga, Japan), were used for production of retrovirus. Helper vectors for lentivirus production, psPAX2 and pMD2.G, were a gift from Didier Trono (Addgene plasmid # 12260; http://n2t.net/addgene:12260; accessed on 1 April 2023. RRID:Addgene_12260) (Addgene plasmid # 12259; http://n2t.net/addgene:12259; accessed on 1 April 2023. RRID:Addgene_12259).

### 4.3. Cell Culture

The CML-derived cell line K562 and murine pro-B cell line Ba/F3 were purchased from the Riken Cell Bank (Ibaraki, Japan). K562 cells were cultured in RPMI 1640 (Nacalai Tesque) supplemented with 10% fetal bovine serum (FBS) (Biowest, Nuaillé, France), 100 units/mL penicillin (Nacalai Tesque), and 100 μg/mL streptomycin (Nacalai Tesque). Ba/F3 cells were cultured in RPMI 1640 supplemented with 10% FBS, 100 units/mL penicillin, 100 μg/mL streptomycin, and IL-3 (2 ng/mL).

### 4.4. Virus Production and Retrovirus Infection

HEK293T cells were transfected with helper virus plasmid together with retroviral plasmid or lentiviral plasmid by FUGENE6 Transfection Reagent (Roche Diagnostics, Indianapolis, IN, USA). Retroviruses and lentiviruses were harvested every 4–6 h between 24 and 60 h post-transfection, pooled, and stored on ice. The collected virus solution was passed through a 0.45 µM filter, aliquoted, and stored at −80 °C until use [56]. Ba/F3 cells were infected with retroviruses expressing BCR-ABL or the BCR-ABL T315I mutant using RetroNectin (Takara Bio Inc., Shiga, Japan) according to the manufacturer’s instructions. Briefly, 0.5 mL of 50 μg/mL RetroNectin was added to each well of an untreated 24-well plate (IWAKI, Shizuoka, Japan) and left overnight at 4 °C. After removing the RetroNectin solution, 1 mL of 2% BSA/PBS was added to each well, left standing at room temperature for 30 min, and after blocking, each well was washed with 1 mL PBS. An amount of 2 mL of retrovirus solution was added to each well coated with RetroNectin, and cultured for 6 h at 37 °C in a 5% CO_2_ incubator. After removing the virus solution, 1 mL of Ba/F3 cells (2 × 10^5^/mL) was added and incubated in a 5% CO_2_ incubator at 37 °C for 72 h as previously described [70]. Ba/F3 cells expressing BCR-ABL or the BCR-ABL T315I mutant were selected by culturing in RPMI medium containing 10% FBS, 100 units/mL penicillin, and 100 μg/mL streptomycin.

### 4.5. Generation of DDX5-Knockout Cells

To knockout DDX5 in K562 cells and Ba/F3 cells expressing BCR-ABL or BCR-ABL-T315I mutants, we used the lentiCRISPRv2 puro vector to express Cas9 and guide RNA for human or mouse DDX5 in these cells by lentivirus. Lentivirus infection was performed using RetroNectin (Takara Bio Inc.) using the same procedure as RetroNectin infection. The infected K562 cells and Ba/F3 cells were selected by culturing in RPMI medium containing 10% FBS, 100 units/mL penicillin, 100 μg/mL streptomycin, 2 ng/mL IL-3, and 2 μg/mL puromycin (InVivoGen). It has been confirmed that infected K562 cells and Ba/F3 cells exhibit survival rate and proliferative ability comparable to control cells in the presence of IL-3. After confirming by immunoblotting that the expression of DDX5 was below the detection limit in each DDX5-KO cell, the pooled cells were used in various experiments.

### 4.6. Immunoblotting

Cells were washed with phosphate buffered saline (PBS) and then cultured in RPMI medium containing 1% FBS or 10% FBS without IL-3, or treated with CPT or FL118 in the same mediums for indicated periods. Cell lysates were prepared and the amounts of protein in the lysates were measured using the Bradford protein assay as previously described [52]. After quantitation, equivalent amounts (20 μg) of protein were separated on 10% or 15% sodium dodecyl sulfate–poly-acrylamide gel electrophoresis (SDS-PAGE) gels and then transferred to polyvinylidene fluoride membranes (MilliporeSigma, St Louis, MO, USA). After blocking, PVDF membranes were incubated with the primary antibody (1:1000 dilution) in 5% skim milk or 1% bovine serum albumin (BSA) in PBS with 0.05% of Tween20 (PBS-T) at 4 °C overnight. After washing three times with PBS-T, PVDF membranes were then incubated with HRP-conjugated secondary (1:3000 dilution) in 5% skim milk or 1% BSA in PBS-T at room temperature for 1 h. After washing five times with PBS-T, bands were visualized by ECL substrate (Thermo Fisher Scientific, Waltham, MA, USA) with Amersham™ Imager 600 (GE Healthcare, Chicago, IL, USA). The intensities of all observed bands were quantified using ImageJ software (version 1.8.0) (U.S. NIH, Bethesda, MD, USA).

### 4.7. Reverse Transcription-Polymerase Chain Reaction (RT-PCR)

Cells were washed with PBS and then cultured in RPMI medium containing 1% FBS or 10% FBS without IL-3, or treated with CPT or FL118 in the same mediums for indicated periods. These cells were treated with imatinib. Total RNA was isolated from K562 cells, transduced K562 cells, and transduced Ba/F3 cells using Sepasol-RNA I Super G (Nacalai Tesque) according to the manufacturer’s instructions. cDNAs were synthesized using ReverTra Ace and oligo dT_20_ (TOYOBO, Osaka, Japan). Real-time PCR using Luna Universal qPCR Master Mix (NEW ENGLAND Biolabs, MA, USA) was performed as previously described [54]. PCR primer sequences were as follows: human BIRC5; (forward) 5′-TTCTCAAGGACCACCGCATC-3′ and (reverse) 5′-AAGACATTGCTAAGGGGCCC-3′, human β2 microglobulin; (forward) 5′-CTCACGTCATCCAGCAGAGA-3′ and (reverse) 5′-CGGCAGGCATACTCATCTTT-3′, mouse birc5; (forward) 5′-CCAGATCTGGCAGCTGTACC-3′ and (reverse) 5′-TGGCTCTCTGTCTGTCCAGT-3′, mouse β2 microglobulin; (forward) 5′-CTGACCGGCCTGTATGCTAT-3′ and (reverse) 5′-TCACATGTCTCGATCCCAGT-3′.

### 4.8. Measurement of Cell Proliferation by Water-Soluble Tetrazolium (WST) Assay

K562 cells and DDX5-KO cells (2 × 10^4^ cells/100 μL or 1 × 10^4^ cells/100 μL) were seeded on 96-well plates using RPMI medium containing 1% or 10% FBS without IL-3. These cells were treated with imatinib (0.125, 0.25, 0.5, 1, and 2 μM), CPT (15.6, 31.3, 62.5, 125, 250, 500, and 1000 nM), or FL118 (15.6, 31.3, 62.5, 125, 250, 500, and 1000 nM), and cultured at 37 °C for 24 h or 48 h. Ba/F3 cells expressing BCR-ABL or the T315I mutant and their DDX5-KO cells (1 × 10^4^ cells/100 μL) were seeded on 96-well plates using RPMI medium containing 1% or 10% FBS. These cells were treated with imatinib (0.06, 0.125, 0.25, 0.5, 1, and 2 μM), CPT (6.25, 12.5, 25, 50, 100, 200, and 400 nM), or FL118 (6.25, 12.5, 25, 50, 100, 200, and 400 nM) and cultured at 37 °C for 24 h. Ten microliters of WST reagent (Nacalai Tesque) was added and cells were then incubated at 37 °C for 2 h. Absorbance at 450/690 nm was measured using the Tecan Infinite multimode (Männedorf, Switzerland). The proliferative ability of each cell was calculated based on the absorbance of untreated cells as 100%.

### 4.9. Measurement of Cell Viability

K562 cells (4 × 10^6^ cells/4 mL) and transduced K562 cells (1 × 10^5^ cells/1 mL) were seeded on a 6 cm dish or 24-well plate using RPMI medium containing 1% or 10% FBS without IL-3, respectively. These cells were treated with imatinib (1 and 2 μM), CPT (125, 250, and 500 nM), or FL118 (125, 250, and 500 nM), and cultured at 37 °C for 24 h or 48 h. Ba/F3 cells expressing BCR-ABL or the BCR-ABL T315I mutant and their DDX5-KO cells (5 × 10^5^ cells/1 mL or 1 × 10^5^ cells/1 mL) were seeded on 24-well plates using RPMI medium containing 1% or 10% FBS without IL-3. These cells were treated with imatinib (2 μM), CPT (50 and 100 nM), or FL118 (50 and 100 nM), and cultured at 37 °C for 18 h or 24 h. Viability was calculated by Trypan blue staining using a Beckman Coulter Vi-Cell (Beckman Coulter, Inc., Brea, CA, USA) as previously reported [52].

### 4.10. Measurement of Topo I Activity

In vitro Topo I activity was measured using the Topoisomerase I assay kit (TopoGEN. Inc., Buena vista, CO, USA) according to the manufacturer’s instructions. Briefly, 250 ng pHOT1 DNA (supercoiled DNA), human recombinant Topo I (1.67 U), and CPT (125, 250, 500, and 1000 nM) or FL118 (125, 250, 500, and 1000 nM) were mixed and incubated at 37 °C for 30 min. After the reactions were stopped by the addition of the attached stop buffer, each sample was loaded onto a 1% agarose gel. The gel was stained with 0.5 μg/mL ethidium bromide and observed under UV irradiation.

### 4.11. Statistical Analysis

Each experiment was performed at least three times and all data were expressed as the mean ± standard deviation. A one-way analysis of variance (ANOVA) was used to evaluate the significance of differences with Prism 8.0.1 (GraphPad, San Diego, CA, USA), and a *p*-value < 0.05 was considered to be significant.

## 5. Conclusions

DDX5 plays an indispensable role in cell proliferation and survival in CML induced by BCR-ABL. The CPT derivative FL118 potently induces apoptosis in CML cells through not only the inhibition of Topo I activity but also the suppression of DDX5 expression. Therefore, FL118 has potential as a novel CML therapeutic agent that may overcome drug resistance to BCR-ABL inhibitors.

## Figures and Tables

**Figure 1 ijms-25-03693-f001:**
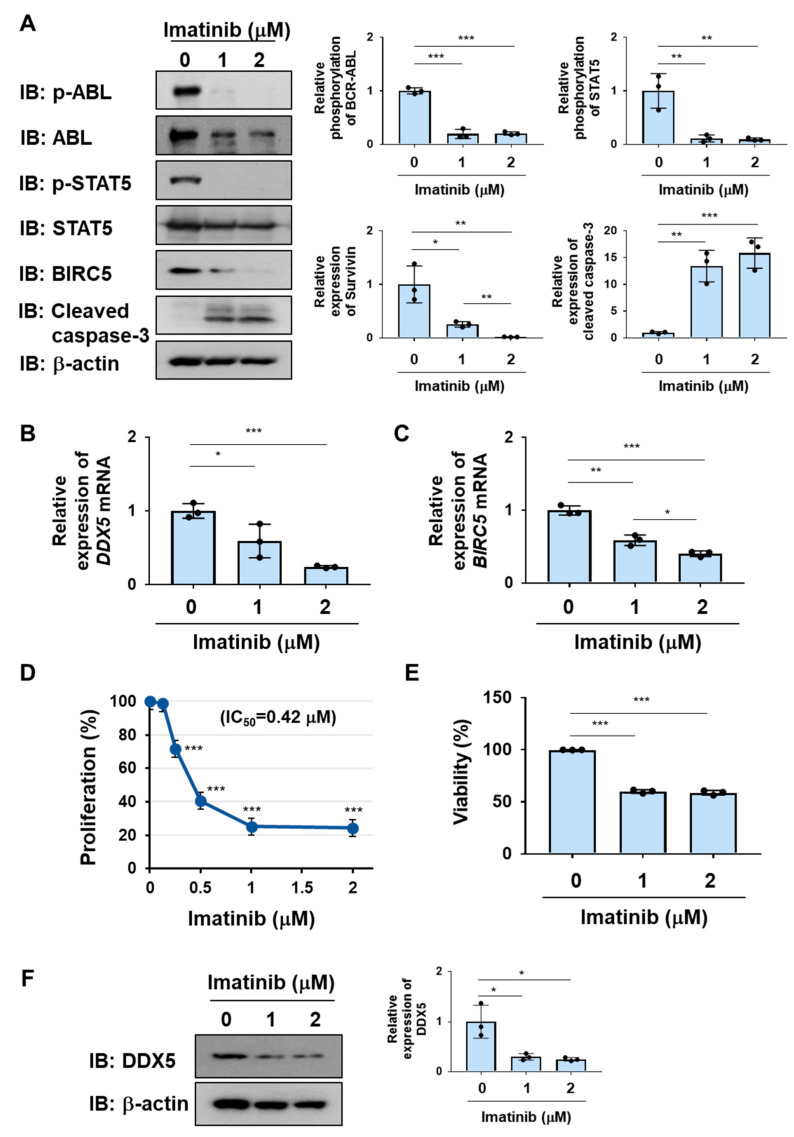
The treatment with imatinib reduces the expression of DDX5 in K562 cells. (**A**,**B**) K562 cells were treated with imatinib (1 and 2 μM) for 24 h. (**A**) The phosphorylation and/or expression levels of BCR-ABL, STAT5, BIRC5, and cleaved caspase-3 were examined by immunoblotting. The relative phosphorylation or expression of each protein is shown in the graphs. *, **, and *** indicate *p* < 0.05, *p* < 0.01, and *p* < 0.001, respectively. (**B**,**C**) The mRNA expression of *DDX5* and *BIRC5* was analyzed by RT-PCR. The mRNA expression level of *β*_2_*-microglobulin* was used as an internal control. *, **, and *** indicate *p* < 0.05, *p* < 0.01 and *p* < 0.001, respectively. (**D**) K562 cells (2 × 10^4^ cells/100 μL) were treated with imatinib (0.125, 0.25, 0.5, 1, and 2 μM) for 48 h. Cell proliferation was assessed by the WST assay. *** indicates *p* < 0.001. (**E**) K562 cells (4 × 10^6^ cells/4 mL) were treated with imatinib (1 and 2 μM) for 48 h. Cell viability was evaluated by the Trypan blue staining method. *** indicates *p* < 0.001. (**F**) The expression level of DDX5 was examined by immunoblotting and the relative expression of DDX5 is shown in the graph. * indicates *p* < 0.05.

**Figure 2 ijms-25-03693-f002:**
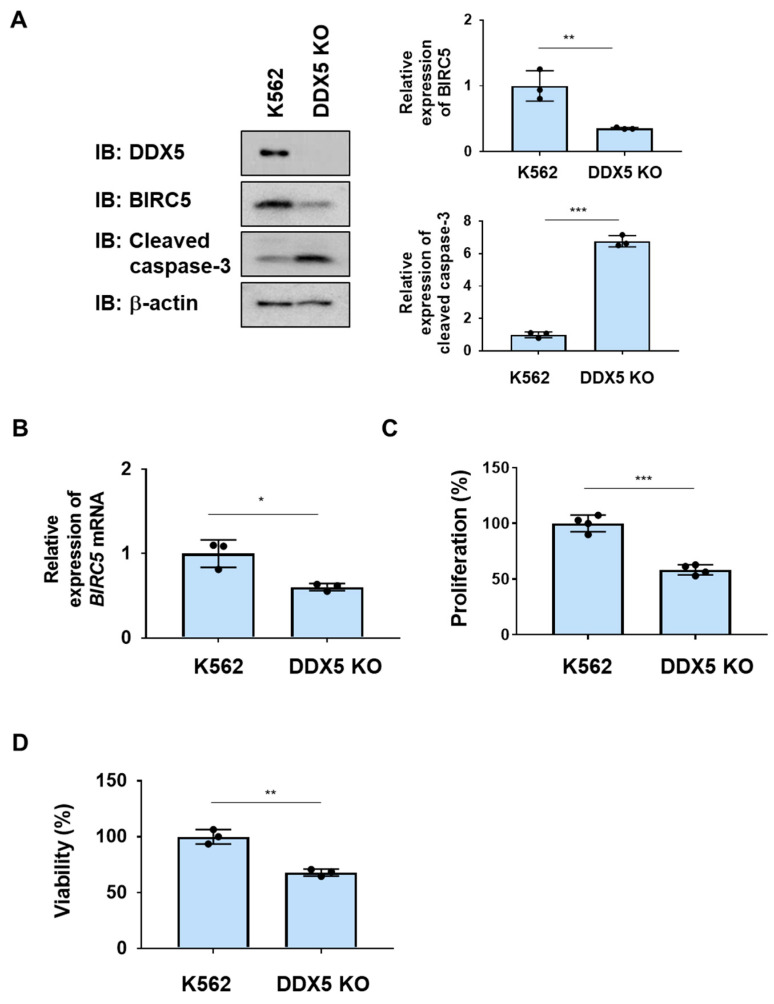
The knockout of DDX5 induces apoptosis in K562 cells. DDX5-KO cells were established by knocking out DDX5 in K562 cells by genome editing. (**A**,**B**) K562 cells and DDX5-KO cells were cultured in RPMI medium containing 1% FBS for 24 h. (**A**) Whole-cell lysates were prepared and the expression of BIRC5, cleaved caspase-3, or β-actin was examined by immunoblotting. The relative expression levels of BIRC5 and cleaved caspase-3 are shown in the graphs. ** and *** indicate *p* < 0.01 and *p* < 0.001, respectively. (**B**) The mRNA expression of BIRC5 was analyzed by RT-PCR. The mRNA expression level of *β_2_-microglobulin* was used as an internal control. * indicates *p* < 0.01. (**C**) Transduced K562 cells (1 × 10^4^ cells/100 μL) were cultured in RPMI containing 1% FBS for 24 h. Cell proliferation was assessed by the WST assay. *** indicates *p* < 0.001. (**D**) Transduced K562 cells (1 × 10^5^ cells/1 mL) were cultured in RPMI containing 1%FBS for 24 h. Cell viability was evaluated by the Trypan blue staining method. ** indicates *p* < 0.01.

**Figure 3 ijms-25-03693-f003:**
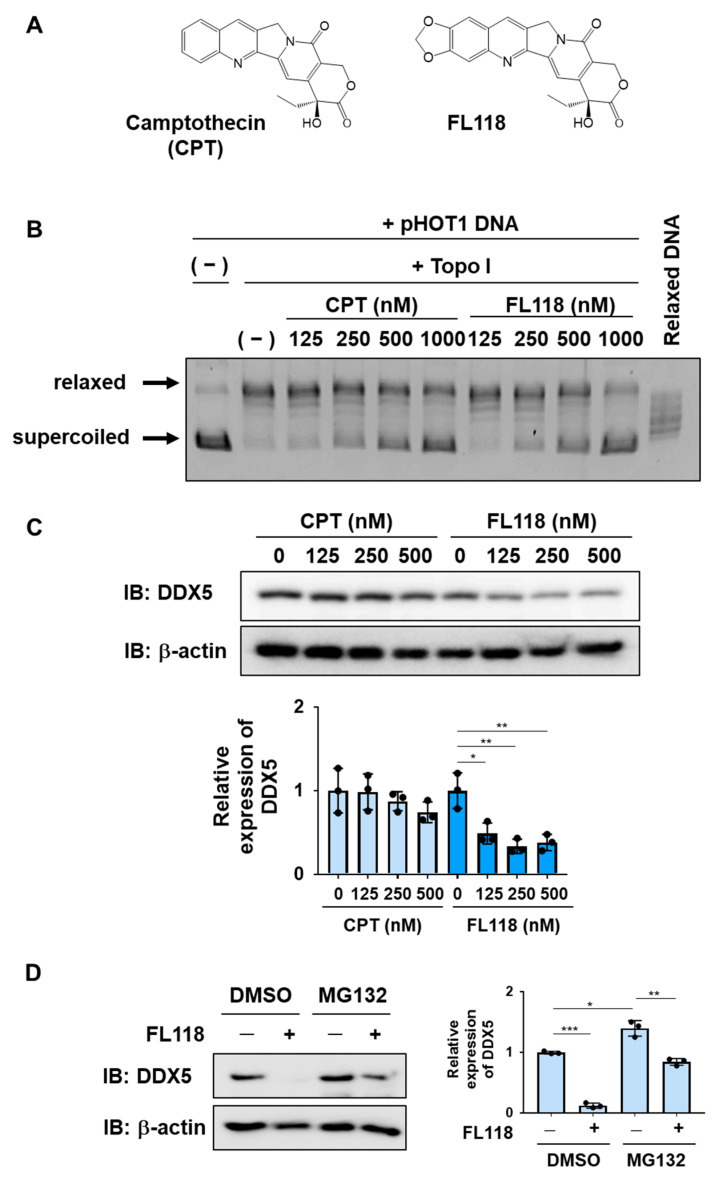
FL118 inhibits the activity of Topo I and suppresses the expression of DDX5. (**A**) The chemical structures of camptothecin (CPT) and FL118 are shown. (**B**) The effects of CPT and FL118 on Topo I activity were measured using recombinant Topo I and supercoiled DNA. After the reaction at 37 °C for 30 min, electrophoresis was performed using a 1% agarose gel. As a control, relaxed DNA was electrophoresed. (**C**) K562 cells were treated with CPT (125, 250, and 500 nM) or FL118 (125, 250, and 500 nM) for 24 h. Whole-cell lysates were prepared, the expression of DDX5 or β-actin was examined by immunoblotting, and the relative expression level of DDX5 is shown in the graph. * and ** indicate *p* < 0.05 and *p* < 0.01, respectively. (**D**) K562 cells were treated with FL118 (500 nM and DMSO (0.1%) or MG-132 (5 μM) for 12 h. Whole-cell lysates were prepared, the expression of DDX5 or β-actin was examined by immunoblotting, and the relative expression level of DDX5 is shown in the graph. *, **, and *** indicate *p* < 0.05, *p* < 0.01, and *p* < 0.001, respectively.

**Figure 4 ijms-25-03693-f004:**
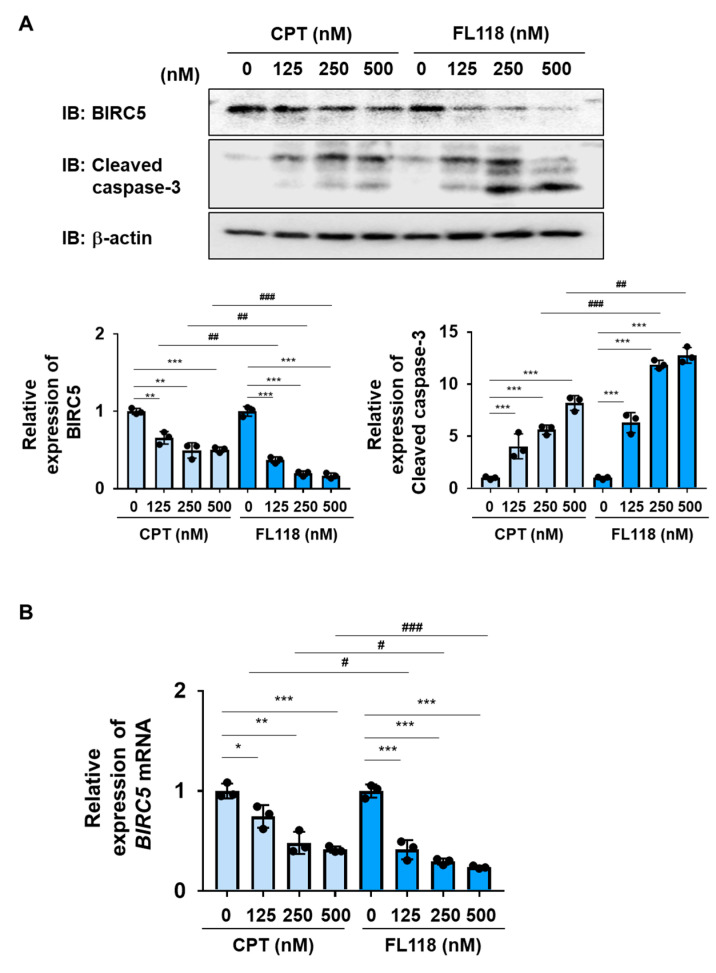
FL118 reduces the expression of BIRC5 and activates caspase-3 in K562 cells. (**A**,**B**) K562 cells were treated with CPT or FL118 at the indicated concentrations for 24 h. (**A**) Whole-cell lysates were prepared and the expression of BIRC5, cleaved caspase-3, or β-actin was examined by immunoblotting. The relative expression level of BIRC5 and cleaved caspase-3 is shown in the graphs. ** and *** indicate *p* < 0.01 and *p* < 0.001, respectively. ^##^ and ^###^ indicate *p* < 0.01 and *p* < 0.001, respectively. (**B**) The mRNA expression of *BIRC5* was analyzed by RT-PCR. The mRNA expression level of *β_2_-microglobulin* was used as an internal control. *, **, and *** indicate *p* < 0.05, *p* < 0.01, and *p* < 0.001, respectively. ^#^ and ^###^ indicate *p* < 0.05 and *p* < 0.001, respectively.

**Figure 5 ijms-25-03693-f005:**
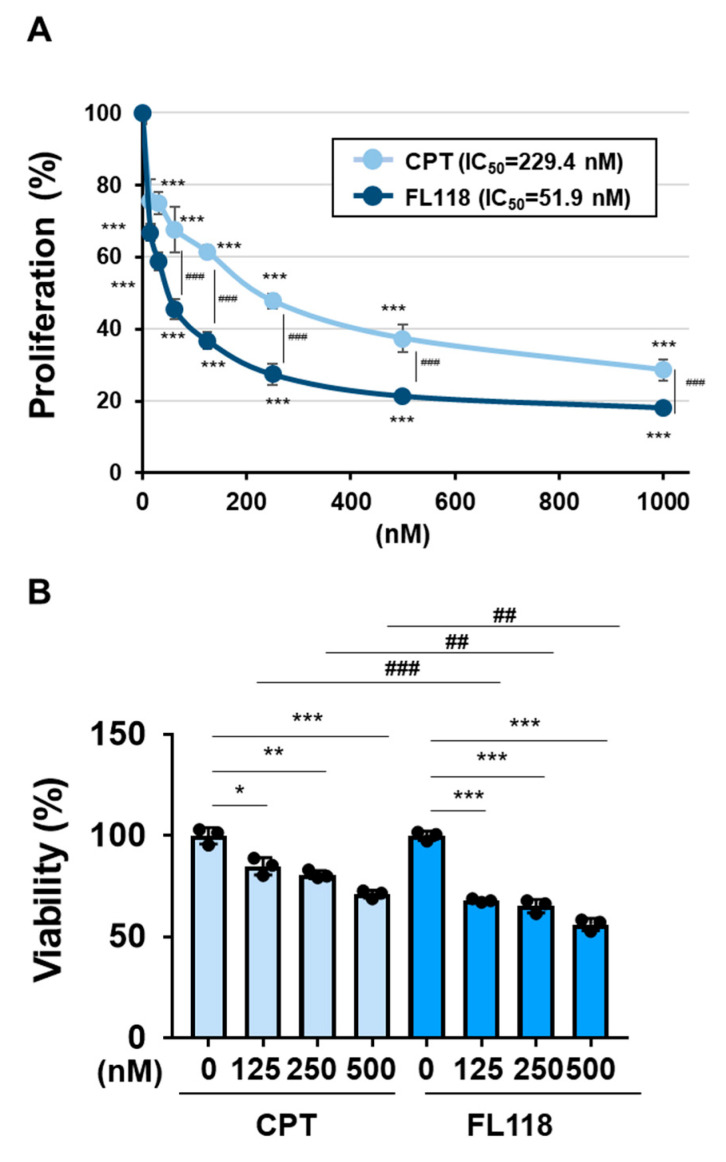
FL118 reduced cell proliferation and induced cell death more strongly than CPT in K562 cells. (**A**) K562 cells (2 × 10^4^ cells/100 μL) were treated with CPT or FL118 at various concentrations for 48 h. Cell proliferation was assessed by WST assay. *** and ^###^ indicate *p* < 0.001. (**B**) Transduced K562 cells (4 × 10^6^ cells/4 mL) were treated with CPT (125, 250, and 500 nM) or FL118 (125, 250, and 500 nM) for 48 h. Cell viability was evaluated by the Trypan blue staining method. *, **, and *** indicate *p* < 0.05, *p* < 0.01, and *p* < 0.001, respectively. ^##^ and ^###^ indicate *p* < 0.01 and *p* < 0.001, respectively.

**Figure 6 ijms-25-03693-f006:**
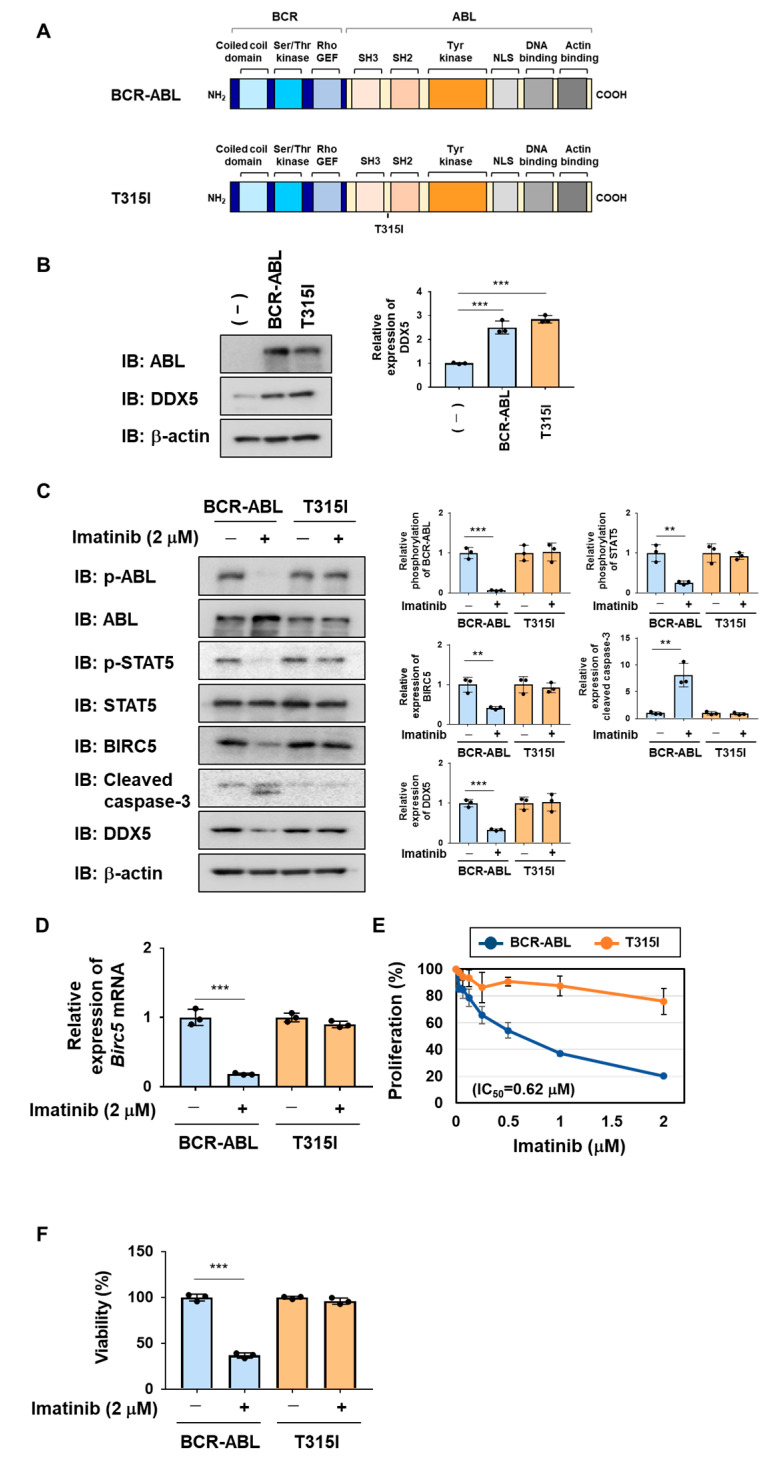
Exogenous expression of BCR-ABL and its T315I mutant induces the expression of DDX5 in Ba/F3 cells. (**A**) The structures of BCR-ABL and its T315I mutant are shown. (**B**–**F**) Ba/F3 cells expressed BCR-ABL and its T315I mutant by retrovirus infection. Ba/F3 cells expressing BCR-ABL or the T315I mutant were incubated with RPMI containing 10% FBS for 24 h. (**B**) Whole-cell lysates were prepared and immunoblotting was performed. The relative expression level of DDX5 is shown in the graph. *** indicates *p* < 0.001. (**C**) Whole-cell lysates were prepared and immunoblotting was performed. The relative phosphorylation level of BCR-ABL or STAT5 and the relative expression of BIRC5, cleaved caspase-3, or DDX5 are shown in the graphs. ** and *** indicate *p* < 0.01 and *p* < 0.001, respectively. (**D**) The mRNA expression of *Birc5* was analyzed by RT-PCR. The mRNA expression level of *β*_2_*-microglobulin* was used as an internal control. *** indicates *p* < 0.001. (**E**) Ba/F3 cells expressing BCR-ABL or the T315I mutant (1 × 10^4^ cells/100 μL) were treated with imatinib at various concentrations for 24 h. Cell proliferation was assessed by the WST assay. (**F**) Ba/F3 cells expressing BCR-ABL or the T315I mutant (5 × 10^5^ cells/1 mL) were treated with imatinib (2 μM) for 18 h. Cell viability was evaluated by the Trypan blue staining method. *** indicates *p* < 0.001.

**Figure 7 ijms-25-03693-f007:**
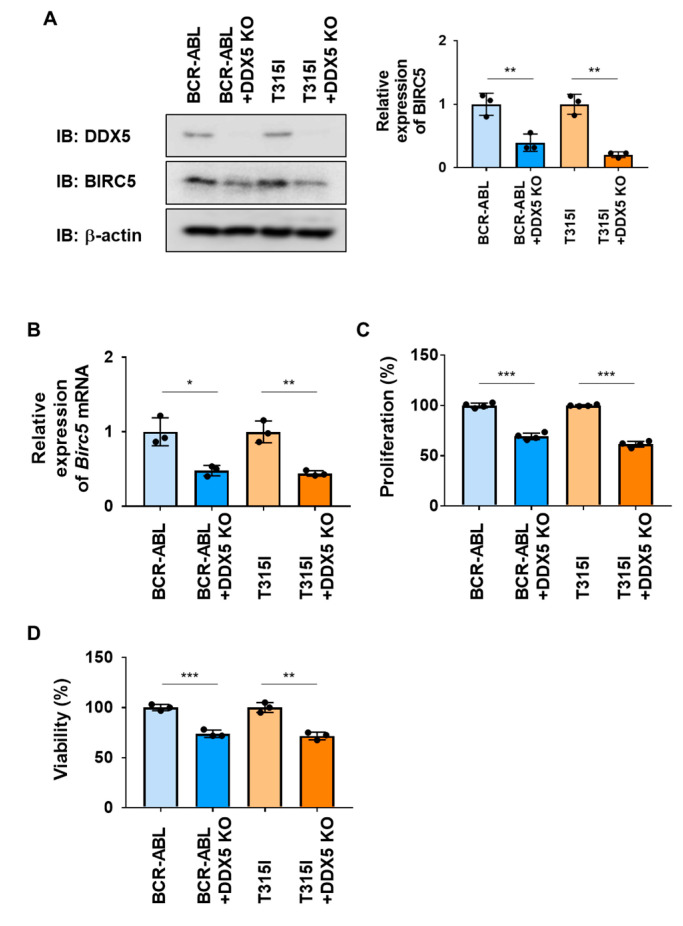
The knockout of DDX5 induces cell death in Ba/F3 cells expressing BCR-ABL or the BCR-ABL T315I mutant. DDX5 was knocked out in Ba/F3 cells expressing BCR-ABL and Ba/F3 cells expressing the BCR-ABL T315I mutant by genome editing. These cells were cultured in RPMI medium containing 10% FBS for 24 h. (**A**) Whole-cell lysates were prepared and the expression of DDX5, BIRC5, or β-actin was examined by immunoblotting. The relative expression levels of BIRC5 are shown in the graph. ** indicates *p* < 0.01. (**B**) The mRNA expression of Birc5 was analyzed by RT-PCR. The mRNA expression level of *β*_2_*-microglobulin* was used as an internal control. * and ** indicate *p* < 0.05 and *p* < 0.01, respectively. (**C**) Transduced Ba/F3 cells (1 × 10^4^ cells/100 μL) were cultured in RPMI containing 1% FBS for 24 h. Cell proliferation was assessed by the WST assay. *** indicates *p* < 0.001. (**D**) Transduced Ba/F3 cells (1 × 10^5^ cells/1 mL) were cultured in RPMI containing 1% FBS for 24 h. Cell viability was evaluated by the Trypan blue staining method. ** and *** indicate *p* < 0.01 and *p* < 0.001, respectively.

**Figure 8 ijms-25-03693-f008:**
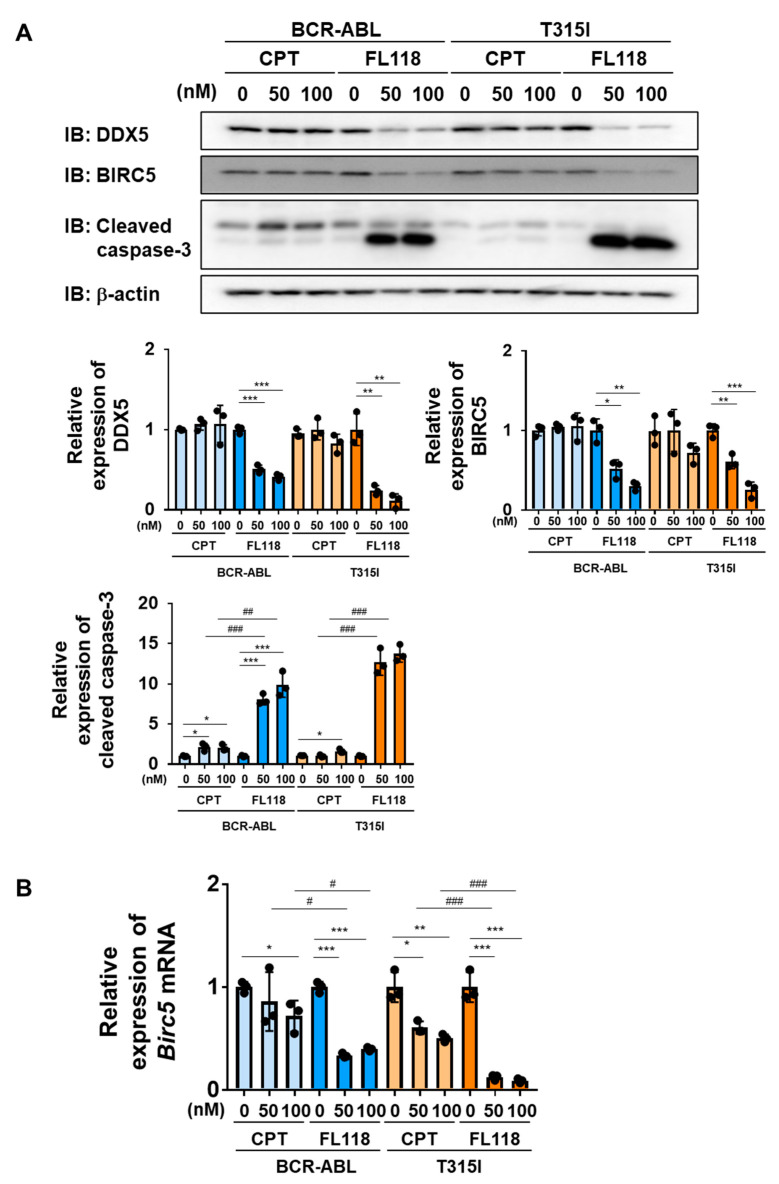
FL118 reduces the expression of BIRC5 and activates caspase-3 in Ba/F3 cells expressing BCR-ABL or the T315I mutant. (**A**,**B**) Ba/F3 cells expressing BCR-ABL or the T315I mutant were treated with CPT (50 and 100 nM) or FL118 (50 and 100 nM) for 24 h. (**A**) Whole-cell lysates were prepared and immunoblotting was performed. The relative expression levels of DDX5, BIRC5, and cleaved caspase-3 are shown in the graphs. *, **, and *** indicate *p* < 0.05, *p* < 0.01, and *p* < 0.001, respectively. ^##^ and ^###^ indicate *p* < 0.01 and *p* < 0.001, respectively. (**B**) The mRNA expression of *Birc5* was analyzed by RT-PCR. The mRNA expression level of *β*_2_*-microglobulin* was used as an internal control. *, **, and *** indicate *p* < 0.05, *p* < 0.01, and *p* < 0.001, respectively. ^#^ and ^###^ indicate *p* < 0.05 and *p* < 0.001, respectively.

**Figure 9 ijms-25-03693-f009:**
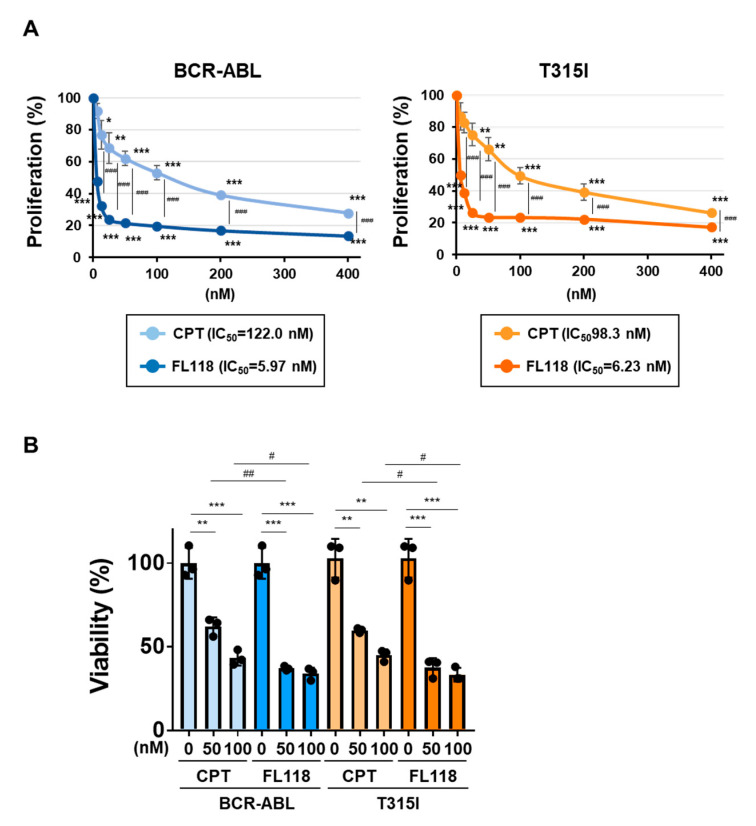
FL118 induces apoptosis in Ba/F3 cells expressing BCR-ABL or the T315I mutant. (**A**) Ba/F3 cells expressing BCR-ABL or the T315I mutant (1 × 10^4^ cells/100 μL) were treated at various concentrations for 24 h. Cell proliferation was assessed by the WST assay. *, **, and *** indicate *p* < 0.05, *p* < 0.01, and *p* < 0.001, respectively. ^###^ indicates *p* < 0.001. (**B**) Ba/F3 cells expressing BCR-ABL or the T315I mutant (5 × 10^5^ cells/1 mL) were treated with CPT (50 and 100 nM) or FL118 (50 and 100 nM) for 18 h. Cell viability was evaluated by the Trypan blue staining method. **, and *** indicate *p* < 0.01, and *p* < 0.001, respectively. ^#^ and ^##^ indicate *p* < 0.05, and *p* < 0.001, respectively.

**Figure 10 ijms-25-03693-f010:**
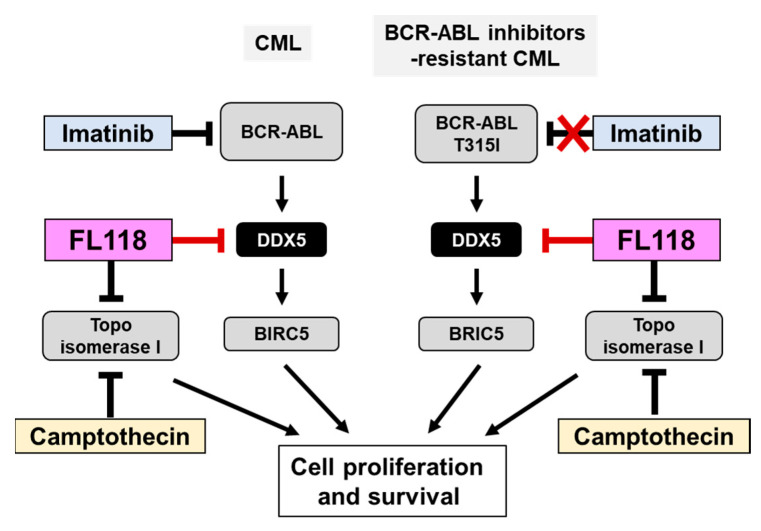
FL118, a DDX5 inhibitor, may be a novel therapeutic agent for CML that is resistant to BCR-ABL inhibitors. BCR-ABL promotes the proliferation and survival of CML cells by inducing the expression of DDX5. Due to the appearance of a point mutation (T315) in BCR-ABL during treatment with the BCR-ABL inhibitor imatinib, CML cells acquire resistance to BCR-ABL inhibitors. Similar to camptothecin (CPT), FL118 inhibits Topo I, but also suppresses the expression of DDX5. Therefore, FL118 has potential as a new treatment for CML that overcomes the drug resistance of CML cells expressing the BCR-ABL T315I mutation.

## Data Availability

Data that support the present results are available from the corresponding author upon reasonable request.

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
