# Peer review of "FL118 Is a Potent Therapeutic Agent against Chronic Myeloid Leukemia Resistant to BCR-ABL Inhibitors through Targeting RNA Helicase DDX5"

_ijms, 2024, doi:10.3390/ijms25073693_

Round 1
Reviewer 1 Report
Comments and Suggestions for Authors
The manuscript by Takeda et al. provides strong evidence supporting the notion that DDX5 is a crucial therapeutic target in CML. Additionally, it highlights the potential of FL118 as a promising candidate compound for treating BCR-ABL inhibitor-resistant CML.
The experiments are well designed, and the manuscript is very nicely written. I have a few suggestions for polishing the manuscript.
Major Comments:
1. The authors should comment whether the application of FL118 affects the RNA helicase activity of DDX5.
2. Experiments suggest that FL118 affects the relative expression of DDX5 proteins. But what is the reason behind this event. The authors should comment whether the relative stability of DDX5 proteins or the translation of DDX5 proteins is altered.
3. The authors commented that DDX5 phosphorylation has an important role in the progression of different cancers. This is indeed true. Is there any evidence that pointed out that post translational modification of DDX5 has a role here as well.
4. The authors should highlight the importance of DDX5 and Survivin interaction more elaborately.
Minor Comments:
1. In the Introduction section the authors should add some references that highlights the role of DDX5 towards the advancement of various cancers.
2. In Figure 4A, the image of Cleaved Caspase 3 is a bit blurry. The authors should change it with a better image.
3. What do authors mean by genome editing methods? Name the method.
4. In Figure 8A, the authors should crop the image of Cleaved Caspase 3 properly.
5. The dilutions of Abs should be mentioned in the Materials and Methods Section.
Author Response
Following the comments and suggestions by the editor and the reviewers, we amended the manuscript. We performed additional experiments and added novel data in Figure 1B, Figure 3D, and Figure 6B in the revised manuscript.
We marked all changes, alternations and additions with red color in the revised manuscript.
Now, we are positive that the reviewer’s comments enabled us to greatly improve our manuscript.
- Comments and Suggestions for Authors
The manuscript by Takeda et al. provides strong evidence supporting the notion that DDX5 is a crucial therapeutic target in CML. Additionally, it highlights the potential of FL118 as a promising candidate compound for treating BCR-ABL inhibitor-resistant CML.
The experiments are well designed, and the manuscript is very nicely written. I have a few suggestions for polishing the manuscript.
Major Comments:
- The authors should comment whether the application of FL118 affects the RNA helicase activity of DDX5.
Answer:
We agreed with this reviewer's valuable comments and added the following sentences in the revised manuscript. (p.18)
“Our observations failed to clarify whether FL118 could affect the RNA helicase activ-ity of DDX5 but the effect of FL118 on the RNA helicase activity should be clarified in near future analysis. To clarify it, we need to purify the active recombinant protein of DDX5 and perform in vitro RNA helicase assay. In the current study, we showed that the ex-pression of DDX5 protein decreased when treated with FL118. Furthermore, the knockout of DDX5 showed comparable inhibitory effects on cell proliferation and BIRC5 expression in both K562 cells and the Ba/F3 cells expressing BCR-ABL as did the treatment with FL118. Therefore, it is thought that the effect of FL118 was more likely to be due to the degradation of DDX5 than inhibition of the helicase activity of DDX5. We also have previous-ly expressed wild-type DDX5 and the DDX5 mutant (K144N) lacking RNA helicase activity in adipocyte precursor 3T3-L1 cells and observed that their expression levels were al-most the same [58]. These results suggested that there is no correlation between the RNA helicase activity of DDX5 and the regulation of its expression.”
- Experiments suggest that FL118 affects the relative expression of DDX5 proteins. But what is the reason behind this event. The authors should comment whether the relative stability of DDX5 proteins or the translation of DDX5 proteins is altered.
Answer:
We would like to thank the reviewers for their valuable opinions. In this revised manuscript, we investigated the effect of the proteasome inhibitor MG132 on the FL118-induced decrease in DDX5 expression, and added the results in Figure 3D and the following sentences in the revised manuscript. (p.3-4, p. 18)
“According to previous report by Ling and colleagues, DDX5 was identified as FL118 binding protein, and FL118 behaves as a molecular glue degrader against DDX5 [70]. This report suggests us the FL118-caused downregulation of DDX5 most unlikely due to deceleration of protein synthesis of DDX5. In fact, we observed that the FL118-induced decrease in DDX5 expression was significantly restored by MG132 treatment in K562 cells (Figure 3D). Therefore, FL118 may promote proteasomal degradation of DDX5 in K562 cells. The detailed mechanism how DDX5 is downregulated by the treatment with FL118 will be a problem which should be clarified in next future project.”
<Reference>
- Ling, X.; Wu, W.; Aljahdali, I.A.M.; Liao, J.; Santha, S.; Fountzilas, C.; Boland, P.M.; Li, FL118, acting as a 'molecular glue degrader', binds to dephosphorylates and degrades the oncoprotein DDX5 (p68) to control c-Myc, survivin and mutant Kras against colorectal and pancreatic cancer with high efficacy. Clin Transl Med. 2022, 12, e881.
- The authors commented that DDX5 phosphorylation has an important role in the progression of different cancers. This is indeed true. Is there any evidence that pointed out that post translational modification of DDX5 has a role here as well.
Answer:
We completely agree with the comment by reviewer. In several studies, it was reported that the activity of DDX5 is regulated by its phosphorylation at Y593, which is catalyzed by c-Abl tyrosine kinase. We additionally described following sentences in discussion in the revised manuscript (p. 18).
“Although we attempted to observe the tyrosine phosphorylation of DDX5 at Y593 in K562 cells by immunoblotting, we failed to see it. It is unclear whether its phosphorylation at Y593 is caused very low level or other tyrosine phosphorylation of DDX5 functions to regulate the activity of DDX5.”
“In the current study, we could conclude that FL118 could be utilized for the treatment of CML resistant to BCR-ABL inhibitor. The functional roles of DDX5 phosphorylation should be clarified in future project.”
- 4. The authors should highlight the importance of DDX5 and Survivin interaction more elaborately.
Answer:
We completely agree with the comments by reviewers. In this revised manuscript, as the reviewer pointed out at the end of the minor points, we used the alias BIRC5 instead of survivin. We described following sentences to emphasize the importance of the relationship between DDX5 and Survivin (BIRC5) (p.18).
“In the current study, we analyzed the expression of BIRC5 as a marker to evaluate the inhibitory effect of FL118 on DDX5. Originally, the functional relationship between DDX5 and BIRC5 was reported by Ling and colleagues [41]. They found that BIRC5 expression was attenuated when the expression of DDX5 was silenced. As described above, it has been well established that DDX5 interacts with various transcription factors and regulates their activity, therefore there is no doubt for considering the involvement of DDX5 in the transcriptional regulation of BIRC5 promoter. This issue will be advanced in the future as the molecular mechanisms of BIRC5 promoter regulation are elucidated.”
Minor Comments:
- In the Introduction section the authors should add some references that highlights the role of DDX5 towards the advancement of various cancers.
Answer:
We completely agree with the comment by reviewer, therefore we additionally described following sentences and cited additional three references in introduction (p.2).
“Iyer and colleagues found that combination of DDX5 and Polo-like kinase-1 (PLK1) positive cases in p53 mutant breast cancer exhibited poor prognosis [32]. Dai et al. reported that high expression of DDX5 and Fructose-bisphosphate aldolase A (ALDOA) are associated with poor prognosis in colorectal cancer (CRC) patients [33]. In addition, it was reported that concomitant long non-coding RNA (lncRNA) NEAT1 and DDX5 protein levels negatively correlated with the survival of CRC patients [34]. We also observed the correlated high expression of DDX5 and a transcription factor c-Myc in CRC patients, and found the presence of their oncogenic positive feedback loop [35]. According to these reports, despite the lack of detailed molecular mechanisms, the importance of DDX5 in cancer development and malignant transformation has been demonstrated by many researchers.”
<References>
- Iyer, R.S.; Nicol, S.M.; Quinlan, P.R. Thompson AM, Meek DW, Fuller-Pace FV. The RNA helicase/transcriptional co-regulator, p68 (DDX5), stimulates expression of oncogenic protein kinase, Polo-like kinase-1 (PLK1), and is associated with elevated PLK1 levels in human breast cancers. Cell Cycle. 2014, 13, 1413-14
- Dai, L.; Pan, G.; Liu, X.; Huang, J.; Jiang, Z.; Zhu, X.; Gan, X.; Xu, Q.; Tan, High expression of ALDOA and DDX5 are associated with poor prognosis in human colorectal cancer. Cancer Manag Res. 2018, 29, 10, 1799-1806.
- Zhang, M.; Weng, W.; Zhang, Q.; Wu, Y; Ni, S.; Tan, The lncRNA NEAT1 activates Wnt/β-catenin signaling and promotes colorectal cancer progression via interacting with DDX5. J Hematol Oncol. 2018, 11, 113.
- Tago, K.; Funakoshi-Tago, M.; Itoh, H.; Furukawa, Y.; Kikuchi, J.; Kato, T.; Suzuki, K.; Yanagisawa, K. Arf tumor suppressor disrupts the oncogenic positive feedback loop including c-Myc and DDX5. Oncogene. 2015, 34, 314-322.
- In Figure 4A, the image of Cleaved Caspase 3 is a bit blurry. The authors should change it with a better image.
Answer:
We changed the image of Cleaved Caspase-3 in Figure 4A.
- What do authors mean by genome editing methods? Name the method.
Answer:
We amend “genome editing methods” into “CRISPR/Casp system”. We also created the item 4.4 Generation of DDX5-KO cells in Materials and Methods, and described how to generate DDX5-knockout K562 cells and Ba/F3 cells expressing BCR-ABL or T315I mutant by genome editing using the CRISPR/Cas9 system.
- In Figure 8A, the authors should crop the image of Cleaved Caspase 3 properly.
Answer:
We amended the data of Cleaved Caspase-3 in Figure 8A.
- The dilutions of Abs should be mentioned in the Materials and Methods Section.
Answer:
We described the conditions of dilution of Abs in the revised manuscript (p.16).
Reviewer 2 Report
Comments and Suggestions for Authors
The current manuscript by Takeda K. et al. discusses the role of BCR-ABL induced expression of DDX5 in CML cell line K562. Inhibition of DDX5 by gene KO or inhibition using FL118 resulted in apoptotic cell death of K562 cells. The findings are of merit if authors have put some efforts to study mechanism and establish connections. The reviewer enthusiasm limited due to the following MAJOR concerns:
1) Thenticate report (to assess the amount of wording duplication in the manuscript) shows 62% match. Authors should take consider to cut down the similarity to ~30% or less.
Title:
2) Title is cumbersome and can be modified for more clarity.
Results:
3) Figure-1: Authors used two high conc. of Imatinib (i.e. 1 & 2 uM) where the cells are stagnant and only 50% cells viable. As shown in figure-1, there is no dose-dependent activity was seen with the doses tested. It is recommended to show the phenotype with lower doses along with tested doses to report a dose-dependent activity.
4) 2.1 conclusion (i.e. BCR-ABL1 induce the expression of DDX5 ) not supported by the data provided in Figure-1. Authors should treat and test the effect of Imatinib treatment in CML cell line without BCR-ABL mutation.
5) Also, it is not clear whether the regulation of DDX5 by BCR-ABL1 is transcriptional (mRNA level) or is it post-translational.
6) Figure-2: a) provide supplementary data to support homozygous KO clone selection. b) Addition of another KO clone to verify the phenotype. c) KO cells looks like sick and not sure how did cells were selected after KO.
7) How does DDX5 KO cells respond to Imatinib?
8) Does DDX5 KO in other CML cells without BCR-ABL1 mutation would strengthen the said conclusions.
9) Figure-2 indicates that DDX5-KO cells are highly apoptotic with only 70% viable and poorly proliferating. Then, please indicate how did these cells survived and cope up with puro selection.
10) Figure-3 and 4 can be combined. As FL118 already known/proven Topo I inhibitor, Figure-3b is not imp in main figures (can be moved to supp. Or deleted).
11) Also, FL118 is a specific inhibitor of DDX5 by suppressing its expression [38]. Then, what is the new results that authors presented in Figure-3?
12) Authors should consider testing and confirming other apoptotic markers other than just Casp-3 levels in Figure-5.
13) Figire-5 caption should be revised based on the proliferation/survival data. It is not indicating apoptosis measurement as authors stated.
14) BaF3 cells used in Figure-6 are stable or transient? The plasmids that indicated in methods do not carry any viral elements.
15) Same question again, whether those BaF3 cells expressing exogenous plasmids are cell pool or single cell clone? It is highly recommended to add t least another clone to confirm the findings.
16) In figure 6B, it is not clear whether authors detected human or mouse proteins in the mouse BaF3 cells. Definitely not human proteins the mouse cell line. If so, how did authors used same antibodies for detecting those proteins in both human and mouse cell lines. Aren’t the antibodies specific to organism?
17) Figure-7 needs a non-transformed BaF3 cells as control see basal levels of DDX5 before infecting with exogenous plasmids.
18) If FL118 induces proteasomal degradation of DDX5, it should tested whether proteasomal inhibitors rescue the phenotype and benefits the cell survival and proliferation with FL118 treatment.
19) In figure-10, it is not connected how the Topo I inhibition lead to CML cells proliferation.
Methods:
20) Section 4.1 should be supplemented with Cat. No# for all the antibodies used in the study.
21) What is the purpose of Bcr/Abl P190-pLEF plasmid (section 4.2)? I think it is for wild-type BCR-ABL expression in Ba/F3 cells. But, Addgene noted that the insert had nucleotide polymorphism resulting in T117M mutation in the ABL-1. Then, it won’t be considered as WT until we know the consequences with that mutation. Instead, authors can use Bcr/Abl P210-pLEF (Plasmid #38158) for wild-type.
22) Authors indicated in section 4.3 that cells were infected with retroviruses and lentiviruses for the same purpose. Also, selection of cells afterwards is repetitive. Which method was adopted for transduction?
23) Also, how did authors selected DDX5-KO cells is missing in the methods.
24) Also, it is not clear whether authors implicated pool of cells or single-cell clone for the study. It is better to include at least two KO clones to see the phenotypic similarity. Here, authors just used once KO cell/ clone.
25) Please provide primers to characterize and confirm DDX5 KO (may be in supplementary)-is it homozygous or heterozygous? Also, where the deletion or indels in the DDX5 gene locus using guide?
26) Also, supplement section 4.3 with variant of HEK293 cells were used for viral packaging.
27) Bcr/Abl P190-pLEF and its T315I variant plasmids are not viral vectors. How did authors generated retroviruses and transduced to make stable BaF3 cell lines?
Other concerns:
28) The study findings are limited to just ONE BCR-AML positive cell line K562. It is very hard to correlate the results from one cell line to all the BCR-ABL positive cell lines. It is highly encouraged to use other BCR-ABL cell lines such as KYO1, LAMA84, EM2, EM3, BV173, AR230, KU812, and KCL22.
29) The study is just limited to In vitro testing and it would be important to establish how did FL118 treatment and DDX5 KO affect the CNL progression in vivo.
30) Authors trying to establish two points: 1) BCR-ABL1 induces DDX5 expression, and 2) DDX5 KO in BCR-ABL1 carrying cells have apoptotic effects. But, the necessary controls are missing to establish both those point in confidence.
31) Data can be cleaned up to discuss them more focused and limiting lot of quant plots to supplementary or indicating the relative levels underneath the blots.
Best wishes,
Comments on the Quality of English Language
Minor edits are needed.
Author Response
Following the comments and suggestions by the editor and the reviewers, we amended the manuscript. We performed additional experiments and added novel data in Figure 1B, Figure 3D, and Figure 6B in the revised manuscript.
We marked all changes, alternations and additions with red color in the revised manuscript.
Now, we are positive that the reviewer’s comments enabled us to greatly improve our manuscript.
- Comments and Suggestions for Authors
The current manuscript by Takeda K. et al. discusses the role of BCR-ABL induced expression of DDX5 in CML cell line K562. Inhibition of DDX5 by gene KO or inhibition using FL118 resulted in apoptotic cell death of K562 cells. The findings are of merit if authors have put some efforts to study mechanism and establish connections. The reviewer enthusiasm limited due to the following MAJOR concerns:
- Thenticate report (to assess the amount of wording duplication in the manuscript) shows 62% match. Authors should take consider to cut down the similarity to ~30% or less.
Answer:
We revised the entire text to reduce the similarity.
Title:
- Title is cumbersome and can be modified for more clarity.
Answer:
We changed the title as follows.
“FL118 is a potential candidate for the therapeutic agent against chronic myeloid leukemia resistant to BCR-ABL inhibitors through targeting RNA helicase DDX5“
Results:
- Figure-1: Authors used two high conc. of Imatinib (i.e. 1 & 2 uM) where the cells are stagnant and only 50% cells viable. As shown in figure-1, there is no dose-dependent activity was seen with the doses tested. It is recommended to show the phenotype with lower doses along with tested doses to report a dose-dependent activity.
Answer:
As the reviewer pointed out, there was no significant difference in the suppressive effects of 1 µM and 2 µM imatinib on the proliferation ability and viability of K562 cells and on the expression of DDX5 (Figure 1B, 1D, 1E, 1F). On the other hand, a significant difference was observed between the treatment with 1 µM and 2 µM imatinib in suppressing the BIRC5 expression and BIRC5 mRNA (Figure 1A, 1C). We mentioned these results and amend the figure legend in Figure 1 in the revised manuscript (p. 3-4). We also added the following sentences by quoting the reference in Discussion section. (p.17)
“In this study, we first investigated the effects of Imatinib on proliferation, survival, and DDX5 expression in K562 cells. With reference to previous reports [60, 61], we chose two concentrations (1 and 2 µM) of imatinib to test their effects. However, there was no significant difference in the suppressive effects of 1 µM and 2 µM imatinib on them (Figure 1). It was reported that K562 cells are mixture of cell populations exhibiting different sensitivity to imatinib and showed that an imatinib-resistant subpopulation is adherent and has upregulated expression of BCR-ABL [62]. This is thought to be the reason why no clear concentration dependence of the effect of imanitib was observed in Figure 1.”
<References>
- Akgun-Cagliyan, G.; Cort-Donmez, A.; Kilic-Toprak, E.; Altintas, Verbascoside potentiates the effect of tyrosine kinase inhibitors on the induction of apoptosis and oxidative stress via the Abl-mediated MAPK signalling pathway in chronic myeloid leukaemia. Exp Ther Med. 2022 24, 514.
- Uchihara, Y.; Tago, K.; Taguchi, H.; Narukawa, Y.; Kiuchi, F.; Tamura, H.; Funakoshi-Tago, Taxodione induces apoptosis in BCR-ABL-positive cells through ROS generation. Biochem Pharmacol. 2018 154, 357-372.
- Karimiani, E.G.; Marriage, F.; Merritt, A.J.; Burthem, J.; Byers, R.J.; Day, P. Single-cell analysis of K562 cells: an imatinib-resistant subpopulation is adherent and has upregulated expression of BCR-ABL mRNA and protein. Exp Hematol. 2014 42, 183-191.e5.
4) 2.1 conclusion (i.e. BCR-ABL1 induce the expression of DDX5) not supported by the data provided in Figure-1. Authors should treat and test the effect of Imatinib treatment in CML cell line without BCR-ABL mutation.
Answer:
We agree with the reviewer's comments and admit that the conclusion was an overstatement. We changed the subtitle to "2.1. The expression of DDX5 was reduced by the treatment with imatinib in K562 cells." and amend the sentences to “Under treatment conditions with imatinib, the expression of DDX5 was significantly reduced (Figure 1F), suggesting that the kinase activity of BCR-ABL is essential for the expression of DDX5 in K562 cells.” (p.3)
- Also, it is not clear whether the regulation of DDX5 by BCR-ABL1 is transcriptional (mRNA level) or is it post-translational.
Answer:
We analyzed the expression of DDX5 mRNA by RT-PCR. We observed that the expression of DDX5 mRNA was significantly reduced by the treatment with imatinib in K562 cells. Therefore, we conclude that the regulation of DDX5 by BCR-ABL is partially due to the transcriptional upregulation. We added this data in Fig. 1B in the revised manuscript (p.3)
- Figure-2: a) provide supplementary data to support homozygous KO clone selection. b) Addition of another KO clone to verify the phenotype. c) KO cells looks like sick and not sure how did cells were selected after KO.
Answer:
We agree that this reviewer's comment is important. However, we were not able to perform some of the experiments pointed out by the reviser during the period given.
We apologize for the confusion caused by the insufficient explanation of our experimental method. We added these experimental procedures are described in “4.3 Cell culture, 4.4. Virus production and retrovirus infection, and 4.5. Generation of DDX5-knockout cells” in Materials and Methods (p.19-20).
In this study, we expressed Cas9 and gRNA for DDX5 in cells by lentiviral infection, and selected infected cells by culturing with medium supplemented with puromycin and IL-3 as described in “Materials and Methods”. After confirming by immunoblotting that the expression of DDX5 was below the detection limit in each KO cell, the pooled cells were used in various experiments. Furthermore, it has been confirmed that DDX5-KO cells exhibit survival rate and proliferative ability comparable to control cells in the presence of IL-3. The experiments shown in Figures 2 A-D were performed after culturing DDX5-KO cells in medium without IL-3 for 24 hours. Therefore, we thought that we were able to examine the effects of DDX5 KO in these experiments.
- How does DDX5 KO cells respond to Imatinib?
Answer:
In DDX5-KO K562 cells, BCR-ABL is present even in the absence of DDX5, so it is thought that it is possible to react with Imatinib. We observed that the treatment with imatinib still exhibits anti-proliferative effects in DDX5-KO K562 cells. Therefore, it was suggested that there are molecules other than DDX5 that play an important role in proliferation induction by BCR-ABL.
- Does DDX5 KO in other CML cells without BCR-ABL1 mutation would strengthen the said conclusions.
Answer:
Thank you for the valuable comment. However, the references listed below shows that BCR/ABL negative CML is a rare hematologic malignancy with an estimated incidence of 1–2% of BCR/ABL positive CML. BCR/ABL negative CML is associated with mutations in Colony Stimulating Factor 3 Receptor (CSF3R), a point mutation of JAK2 (V617F) or KRAS or NRAS mutations [1, 2]. So far, we have found that DDX5 expression is induced by the JAK2 V617F mutant and plays an important role in proliferation and tumorigenesis [3]. We also report that the expression of DDX5 is induced by the transcription factor Myc activated by KRAS mutant, and that DDX5 plays an essential role in Myc-induced transformation [4]. Therefore, DDX5 is considered to be a factor that commonly functions downstream of oncogene products other than BCR-ABL.
In this study, we found for the first time that DDX5 plays an important role in BCR-ABL-induced cell proliferation and survival using K562 cells and Ba/F3 cells expressing BCR-ABL. Based on these results, we would like to propose that targeting DDX5 could be a novel therapeutic strategy for CML.
<References>
- Conter, C. Rizzari, A. Sala, R. Chiesa, M. Citterio, A. Biondi Acute lymphoblastic leukemia. Orphanet Encyclopedia (2004), pp. 1-13
- Giri S, Pathak R, Martin MG, Bhatt VR. Characteristics and survival of BCR/ABL negative chronic myeloid leukemia: a retrospective analysis of the Surveillance, Epidemiology and End Results database. Ther Adv Hematol. 2015 Dec;6(6):308-12. doi: 10.1177/2040620715607416. PMID: 26622999; PMCID: PMC4649605.
- Takeda K, Tago K, Funakoshi-Tago M. The indispensable role of the RNA helicase DDX5 in tumorigenesis induced by the myeloproliferative neoplasm-associated JAK2V617F mutant. Cell Signal. 2023 Feb;102:110537. doi: 10.1016/j.cellsig.2022.110537. Epub 2022 Nov 26. PMID: 36442590.
- Tago K, Funakoshi-Tago M, Itoh H, Furukawa Y, Kikuchi J, Kato T, Suzuki K, Yanagisawa K. Arf tumor suppressor disrupts the oncogenic positive feedback loop including c-Myc and DDX5. Oncogene. 2015 Jan 15;34(3):314-22. doi: 10.1038/onc.2013.561. Epub 2014 Jan 27. PMID: 24469041.
- Figure-2 indicates that DDX5-KO cells are highly apoptotic with only 70% viable and poorly proliferating. Then, please indicate how did these cells survived and cope up with puro selection.
Answer:
We apologize for the insufficient explanation of the experimental conditions. This is the same answer as the previous comment 5), we used a medium containing IL-3 in addition to puromycin for passage and selection of infected cells, but in the presence of IL-3, and confirmed that the proliferative rate and viability of control and STAT5-KO cells were approximately the same. We stated that each experiment was performed after incubating puromycin resistant cells in medium without IL-3 (p. 20).
- Figure-3 and 4 can be combined. As FL118 already known/proven Topo I inhibitor, Figure-3b is not imp in main figures (can be moved to supp. Or deleted).
Answer:
As pointed out by the reviewer, FL118 has been reported to exhibit TopoI inhibitory activity, but no experiment has actually been performed to date to compare the TopoI inhibitory activity of CPT and that of FL118.
So far, Ling and his/her colleagues have compared the TopoI inhibitory activity of SN-38 and FL118 at only one constant concentration (1 μM) [1]. Therefore, the data in Fig. 3B are new results in terms of comparing the TopoI inhibitory activity of CPT and FL118 at various concentrations. We would like to leave Fig. 3B in the main body of the paper, and we would appreciate your understanding of our opinion.
<Reference>
- Ling X, Cao S, Cheng Q, Keefe JT, Rustum YM, Li F. A novel small molecule FL118 that selectively inhibits survivin, Mcl-1, XIAP and cIAP2 in a p53-independent manner, shows superior antitumor activity. PLoS One. 2012;7(9):e45571. doi: 10.1371/journal.pone.0045571. Epub 2012 Sep 19. PMID: 23029106; PMCID: PMC3446924.
- Also, FL118 is a specific inhibitor of DDX5 by suppressing its expression [38]. Then, what is the new results that authors presented in Figure-3?
Answer:
The new result is that FL118 was confirmed to suppress DDX5 expression also in K562 cells. Therefore, we have deleted the sentence “FL118 is a specific inhibitor of DDX5 by suppressing its expression [38].”
- Authors should consider testing and confirming other apoptotic markers other than just Casp-3 levels in Figure-5.
Answer:
As commented by reviewer, we agree that it would be better to test other apoptotic markers other than caspase-3. Although we additionally observed that the expression of an anti-apoptosis protein Mcl-1 was reduced by the treatment with FL118 in K562 cells, the results are still preliminary. It is well understood that the activation of caspase-3 is as one of most important determinants for the apoptotic cell death, and we chose it as a marker for apoptotic cell death. We appreciate your understanding our opinion.
- Figire-5 caption should be revised based on the proliferation/survival data. It is not indicating apoptosis measurement as authors stated.
Answer:
We amended Figure 5 caption into “FL118 reduced cell proliferation and induced cell death more strongly than CPT in K562 cells.” (p. 9)
14) BaF3 cells used in Figure-6 are stable or transient? The plasmids that indicated in methods do not carry any viral elements.
Answer:
We appreciated for the detailed review. We had incorrectly listed the plasmid information. We amended 4.2. Plasmids in Materilas and methods as following (p. 19).
pSG5-P190 was a gift from Nora Heisterkamp (Addgene plasmid # 31285 ; http://n2t.net/addgene:31285 ; RRID:Addgene_31285). The cDNA encoding p190 BCR-ABL was inserted into the MSCV-IRES-GFP retroviral vector. The mutagenesis of amino acid residues in p190 BCR-ABL (T315I) was performed using a site-directed mutagenesis kit (Stratagene, La Jolla, CA, USA).
15) Same question again, whether those BaF3 cells expressing exogenous plasmids are cell pool or single cell clone? It is highly recommended to add t least another clone to confirm the findings.
Answer:
We added the methods for virus preparation and selection of infected cells using puromycin to "Materials and Methods". Furthermore, we mentioned that pooled infected cells resistant to puromycin were used for each experiment (p. 19-20).s
16) In figure 6B, it is not clear whether authors detected human or mouse proteins in the mouse BaF3 cells. Definitely not human proteins the mouse cell line. If so, how did authors used same antibodies for detecting those proteins in both human and mouse cell lines. Aren’t the antibodies specific to organism?
Answer:
In Figure 6D, the expression and phosphorylation of exogenous human BCR-ABL was detected using anti-phospho c-ABL antibody and anti-c-ABL antibody. These antibodies can detect human and mouse c-ABL, as well as BCR-ABL. In Figure 1A and Figure 6D, we illustrated that we detected BCR-ABL.
Whereas anti-cleaved caspase-3 antibody detects only murine cleaved caspase-3, other antibodies can react with human and mouse each molecule. Therefore, these antibodies detect the expression and phosphorylation of endogenous murine STAT5, DDX5, caspase-3, and β-actin in Ba/F3 cells.
17) Figure-7 needs a non-transformed BaF3 cells as control see basal levels of DDX5 before infecting with exogenous plasmids.
Answer:
We interpreted this as a comment regarding Fig.6. The expression of DDX5 was slightly observed in control Ba/F3 cells and enforced expression of BCR-ABL or its T315I mutant significantly induced the expression of DDX5. The results are shown in Fig. 6B in the revised manuscript (p.10).
18) If FL118 induces proteasomal degradation of DDX5, it should tested whether proteasomal inhibitors rescue the phenotype and benefits the cell survival and proliferation with FL118 treatment.
Answer:
In the revised manuscript, we showed that the proteasome inhibitor MG132 significantly relieves FL118-induced suppression of DDX5 expression in K562 cells in Figure 3D. Therefore, it was suggested that FL118 promotes proteasomal degradation of DDX5. However, it has been reported that the MG132 used this time inhibits all intracellular proteasomes and exhibits potent cytotoxicity [1]. Therefore, even if MG132 suppresses the decrease in DDX5 expression, it is predicted that it will be difficult to restore the viability of K562 cells treated with FL118.
<Reference>
- Guo N, Peng Z. MG132, a proteasome inhibitor, induces apoptosis in tumor cells. Asia Pac J Clin Oncol. 2013 Mar;9(1):6-11. doi: 10.1111/j.1743-7563.2012.01535.x. Epub 2012 May 15. PMID: 22897979.
19) In figure-10, it is not connected how the Topo I inhibition lead to CML cells proliferation.
Answer:
We appreciate the comments by reviewer. We additionally quoted the reference and described following sentences in discussion, and added additional reference. (p.13)
When considering the mechanism of action of FL118, it is necessary to discuss the inhibitory effect of FL118 on Topo I, as the treatment with CPT has been reported to cause apoptosis through activation of caspase-3 [43]. Therefore, it is thought that FL118 exhibits cytotoxicity against CML cells by not only suppressing DDX5 expression but also inhibiting Topo I (Figure 10).
<Reference>
- Kiechle FL, Zhang X. Apoptosis: biochemical aspects and clinical implications. Clin Chim Acta. 2002 Dec;326(1-2):27-45. doi: 10.1016/s0009-8981(02)00297-8.
Methods:
20) Section 4.1 should be supplemented with Cat. No# for all the antibodies used in the study.
Answer:
We added Cat. No# for each antibody (p. 19).
21) What is the purpose of Bcr/Abl P190-pLEF plasmid (section 4.2)? I think it is for wild-type BCR-ABL expression in Ba/F3 cells. But, Addgene noted that the insert had nucleotide polymorphism resulting in T117M mutation in the ABL-1. Then, it won’t be considered as WT until we know the consequences with that mutation. Instead, authors can use Bcr/Abl P210-pLEF (Plasmid #38158) for wild-type.
Answer:
Thank you very much for pointing out this mistake. We corrected plasmid information in the revised manuscript (p. 19).
22) Authors indicated in section 4.3 that cells were infected with retroviruses and lentiviruses for the same purpose. Also, selection of cells afterwards is repetitive. Which method was adopted for transduction?
Answer:
We created items "4.4. Virus production and retrovirus infection" and "4.5. Generation of DDX5-knockout cells" and also described each method in Material and Methods in the revised manuscript (p. 19-20).
23) Also, how did authors selected DDX5-KO cells is missing in the methods.
Answer:
We added the method to select DDX5-KO cells in the revised manuscript (p. 20).
24) Also, it is not clear whether authors implicated pool of cells or single-cell clone for the study. It is better to include at least two KO clones to see the phenotypic similarity. Here, authors just used once KO cell/ clone.
Answer:
In this study, it was confirmed by immunoblotting that the expression of DDX5 was below the detection limit in DDX5-KO cells. Therefore, we used the pooled DDX5-KO cells. We wrote the information of cells used in Material and Methods (p. 20).
25) Please provide primers to characterize and confirm DDX5 KO (may be in supplementary)-is it homozygous or heterozygous? Also, where the deletion or indels in the DDX5 gene locus using guide?
Answer:
In response to the reviewers' other comments, we confirmed the DDX5 deletion by immunoblotting. Additionally, we used the pooled each infected cell to eliminate the effects of clonal variation. We added the information in the revised manuscript.
It seems important to determine whether DDX5 KO is homozygous or heterozygous, but we were not able to do so during this revision period. Although we did not show the data in the manuscript, we observed the comparable results when the expression of DDX5 was silenced by shRNA. Therefore, we conclude that the reduction in proliferation and survival of K562 cells and Ba/F3 cells expressing BCR-ABL is a specific effect of DDX5 deletion.
26) Also, supplement section 4.3 with variant of HEK293 cells were used for viral packaging.
Answer:
We have added information on the virus preparation method using HEK293T cells to Materials and Methods (p.19-20).
27) Bcr/Abl P190-pLEF and its T315I variant plasmids are not viral vectors. How did authors generated retroviruses and transduced to make stable BaF3 cell lines?
Answer:
We apologize for writing the incorrect plasmid information. In the revised manuscript, we corrected the plasmid information and added the methods for production of viruses, infection, and generation of KO cells (p. 19-20).
Other concerns:
28) The study findings are limited to just ONE BCR-AML positive cell line K562. It is very hard to correlate the results from one cell line to all the BCR-ABL positive cell lines. It is highly encouraged to use other BCR-ABL cell lines such as KYO1, LAMA84, EM2, EM3, BV173, AR230, KU812, and KCL22.
Answer:
As the reviewer commented, we only analyzed the function of DDX5 in K562 cells and artificial Ba/F3 cells expressing BCR-ABL. We understand that it is necessary to examine the anti-tumor activity of FL118 using other CML cell lines. Therefore, in the revised manuscript, we added the following sentences in Discussion section (p. 18-19).
“In the absence of IL-3, the expression of DDX5 was slightly observed in Ba/F3 cells, but the enforced expression of BCR-ABL significantly induced DDX5 expression (Figure 6B). Considering this phenomenon and the results obtained with DDX knockout and treatment with FL118, we proposed a molecular mechanim by which enhanced expression of DDX5 positively regulates proliferation and survival of BCR-ABL-positive CML cells (Figure 10). However, we only analyzed the function of her DDX5 in K562 cells and artificial BCR-ABL expressing Ba/F3 cells in this study. In the future study, we need to demonstrate the importance of DDX5 in CML pathogenesis using other BCR-ABL-positive CML cell lines such as KYO1, LAMA84, EM2, EM3, BV173, AR230, KU812, and KCL22. Furthermore, it is necessary to examine the effect of FL118 on CML progression using CML model mice and verify the anti-tumor activity of FL118 in vivo.”
29) The study is just limited to In vitro testing and it would be important to establish how did FL118 treatment and DDX5 KO affect the CNL progression in vivo.
Answer:
We agree with the reviewer's comment and feel that it is important to examine the anti-tumor activity of FL118 in vivo in the future. We have added to the Discussion section that in the future, in vivo analysis will be important in verifying the effectiveness of FL118 as a CML treatment. (p. 20)
30) Authors trying to establish two points: 1) BCR-ABL1 induces DDX5 expression, and 2) DDX5 KO in BCR-ABL1 carrying cells have apoptotic effects. But, the necessary controls are missing to establish both those point in confidence.
Answer:
Regarding 1), we have already responded to the above comment, and we added results that the enforced expression of BCR-ABL induces DDX5 expression in Ba/F3 cells in Figure 6B in the revised manuscript.
Regarding 2), we agree with the reviser’s comment. Both the knockout of DDX5 and the treatment with FL118 treatment similarly induced apoptosis in K562 cells, whereas the treatment with FL118 but not the knockout of DDX5 induced caspase-3 activation in Ba/F3 cells expressing BCR-ABL. We have not yet been able to provide an answer regarding this reason. In the future, we think it will be necessary to change the method of suppressing DDX5 expression and consider other indicators of apoptosis other than caspase-3.
31) Data can be cleaned up to discuss them more focused and limiting lot of quant plots to supplementary or indicating the relative levels underneath the blots.
Answer:
Following the reviewer's comments above, we carefully reviewed the content of the paper. Once we wrote the quantitative values under the blot, however, we felt that the data seemed to be difficult to understand. Therefore, we left the graphs as they are now. Also, since it is necessary to discuss the significant difference in the effects of CPT and FL118, we thought it would be better not to include the quantitative graphs in the supplementary data, so we left them in its current state. We hope the reviewer understand our opinion.
Reviewer 3 Report
Comments and Suggestions for Authors
The manuscript titled “Analysis of FL118 as a potent candidate therapeutic agent for CML resistant to BCR-ABL inhibitors: Functional involvement of DDX5 in BCR-ABL-induced oncogenic signals” investigates the role of DDX5 in chronic myeloid leukemia (CML) cells and its potential as a therapeutic target, especially for BCR-ABL inhibitor resistant cases. The authors use a combination of genetic tools (CRISPR knockout) and pharmacological inhibition (FL118) to demonstrate that DDX5 is critical for CML cell proliferation and survival. They further show efficacy of the DDX5 inhibitor FL118 against imatinib-resistant cells expressing the T315I mutation. The topic is significant, and the approaches are sound. The data overall support the conclusions.
Major comments:
1. While the introduction effectively establishes the background of CML and resistance to BCR-ABL inhibitors, it would benefit from further context specifically focused on DDX5 and its established roles in cancer and proliferation. This additional information would help strengthen the rationale for the study.
2. While Figure 1 demonstrates a statistically significant difference between imatinib treatment and the control group, it does not address the dose response between the 1 μM and 2 μM concentrations. To ensure clarity, please clarify in the text and figure legend whether there is a significant difference between these two doses.
3. Figure 8 shows reduction of survivin mRNA by FL118 but not much change by CPT. However, the protein data in Figure 4 shows CPT also reduces survivin. Please discuss this discrepancy in the results section.
4. Any mechanistic insights on how DDX5 is acting downstream of BCR-ABL? Through transcriptional regulation, effects on RNA processing, etc? Please discuss current knowledge and suggestions for future investigation.
5. Is FL118 binding and degrading DDX5 in this CML system? Figure 3 shows it reduces DDX5 protein but a more explicit confirmation would be helpful.
Minor comments:
1. Abbreviate chronic myeloid leukemia as CML after first use.
2. Standardize capitalization and italicization of gene/protein names throughout manuscript.
Author Response
Following the comments and suggestions by the editor and the reviewers, we amended the manuscript. We performed additional experiments and added novel data in Figure 1B, Figure 3D, and Figure 6B in the revised manuscript.
We marked all changes, alternations and additions with red color in the revised manuscript.
Now, we are positive that the reviewer’s comments enabled us to greatly improve our manuscript.
2. Comments and Suggestions for Authors
The manuscript titled “Analysis of FL118 as a potent candidate therapeutic agent for CML resistant to BCR-ABL inhibitors: Functional involvement of DDX5 in BCR-ABL-induced oncogenic signals” investigates the role of DDX5 in chronic myeloid leukemia (CML) cells and its potential as a therapeutic target, especially for BCR-ABL inhibitor resistant cases. The authors use a combination of genetic tools (CRISPR knockout) and pharmacological inhibition (FL118) to demonstrate that DDX5 is critical for CML cell proliferation and survival. They further show efficacy of the DDX5 inhibitor FL118 against imatinib-resistant cells expressing the T315I mutation. The topic is significant, and the approaches are sound. The data overall support the conclusions.
Major comments:
- While the introduction effectively establishes the background of CML and resistance to BCR-ABL inhibitors, it would benefit from further context specifically focused on DDX5 and its established roles in cancer and proliferation. This additional information would help strengthen the rationale for the study.
Answer:
This has been also pointed out by another reviewer, and we agree with it is important mentioning the relationship between DDX5 and various cancers in Introduction. We described following sentences and cited additional three references in introduction (p.2).
“Iyer and colleagues found that combination of DDX5 and Polo-like kinase-1 (PLK1) positive cases in p53 mutant breast cancer exhibited poor prognosis [32]. Dai et al. reported that high expression of DDX5 and Fructose-bisphosphate aldolase A (ALDOA) are associated with poor prognosis in colorectal cancer (CRC) patients [33]. In addition, it was reported that concomitant long non-coding RNA (lncRNA) NEAT1 and DDX5 protein levels negatively correlated with the survival of CRC patients [34]. We also observed the correlated high expression of DDX5 and a transcription factor c-Myc in CRC patients, and found the presence of their oncogenic positive feedback loop [35]. According to these reports, despite the lack of detailed molecular mechanisms, the importance of DDX5 in cancer development and malignant transformation has been demonstrated by many researchers.”
<References>
- Iyer, R.S.; Nicol, S.M.; Quinlan, P.R. Thompson AM, Meek DW, Fuller-Pace FV. The RNA helicase/transcriptional co-regulator, p68 (DDX5), stimulates expression of oncogenic protein kinase, Polo-like kinase-1 (PLK1), and is associated with elevated PLK1 levels in human breast cancers. Cell Cycle. 2014, 13, 1413-14
- Dai, L.; Pan, G.; Liu, X.; Huang, J.; Jiang, Z.; Zhu, X.; Gan, X.; Xu, Q.; Tan, High expression of ALDOA and DDX5 are associated with poor prognosis in human colorectal cancer. Cancer Manag Res. 2018, 29, 10, 1799-1806.
- Zhang, M.; Weng, W.; Zhang, Q.; Wu, Y; Ni, S.; Tan, The lncRNA NEAT1 activates Wnt/β-catenin signaling and promotes colorectal cancer progression via interacting with DDX5. J Hematol Oncol. 2018, 11, 113.
- Tago, K.; Funakoshi-Tago, M.; Itoh, H.; Furukawa, Y.; Kikuchi, J.; Kato, T.; Suzuki, K.; Yanagisawa, K. Arf tumor suppressor disrupts the oncogenic positive feedback loop including c-Myc and DDX5. Oncogene. 2015, 34, 314-322.
- While Figure 1 demonstrates a statistically significant difference between imatinib treatment and the control group, it does not address the dose response between the 1 μM and 2 μM concentrations. To ensure clarity, please clarify in the text and figure legend whether there is a significant difference between these two doses.
Answer:
As the reviewer pointed out, there was no significant difference in the suppressive effects of 1 µM and 2 µM imatinib on the proliferation ability and viability of K562 cells and on the expression of DDX5 (Figure 1B, 1D, 1E, 1F). On the other hand, a significant difference was observed between the treatment with 1 µM and 2 µM imatinib in suppressing the BIRC5 expression and BIRC5 mRNA (Figure 1A, 1C). We mentioned these results and amend the figure legend in Figure 1 in the revised manuscript (p. 3-4). We also added the following sentences by quoting the reference in Discussion section. (p.17)
“In this study, we first investigated the effects of Imatinib on proliferation, survival, and DDX5 expression in K562 cells. With reference to previous reports [60, 61], we chose two concentrations (1 and 2 µM) of imatinib to test their effects. However, there was no significant difference in the suppressive effects of 1 µM and 2 µM imatinib on them (Figure 1). It was reported that K562 cells are mixture of cell populations exhibiting different sensitivity to imatinib and showed that an imatinib-resistant subpopulation is adherent and has upregulated expression of BCR-ABL [62]. This is thought to be the reason why no clear concentration dependence of the effect of imanitib was observed in Figure 1.”
<References>
- Akgun-Cagliyan, G.; Cort-Donmez, A.; Kilic-Toprak, E.; Altintas, Verbascoside potentiates the effect of tyrosine kinase inhibitors on the induction of apoptosis and oxidative stress via the Abl-mediated MAPK signalling pathway in chronic myeloid leukaemia. Exp Ther Med. 2022 24, 514.
- Uchihara, Y.; Tago, K.; Taguchi, H.; Narukawa, Y.; Kiuchi, F.; Tamura, H.; Funakoshi-Tago, Taxodione induces apoptosis in BCR-ABL-positive cells through ROS generation. Biochem Pharmacol. 2018 154, 357-372.
- Karimiani, E.G.; Marriage, F.; Merritt, A.J.; Burthem, J.; Byers, R.J.; Day, P. Single-cell analysis of K562 cells: an imatinib-resistant subpopulation is adherent and has upregulated expression of BCR-ABL mRNA and protein. Exp Hematol. 2014 42, 183-191.e5.
- Figure 8 shows reduction of survivin mRNA by FL118 but not much change by CPT. However, the protein data in Figure 4 shows CPT also reduces survivin. Please discuss this discrepancy in the results section.
Answer:
As the reviewer pointed out, there were differences in the effects of CPT on the protein and mRNA expression of surviving (BIRC5) in K562 cells and Ba/F3 cells expressing BCR-ABL. We quoted the additional five references and added the following sentences in the revised manuscript (p.17-18).
“In this study, we examined the anti-tumor activiy of FL118 and CPT against K562 cells and Ba/F3 cells expressing BCR-ABL or its T315I mutant (Figures 4 and 8). Through the experiments, both common and different effects of FL118 and CPT were found, and some of them were discrepant results that are difficult to explain the circumstances. The treatment with CPT suppressed the expression of BIRC5 mRNA and BIRC5 protein in K562 cells, although it was weaker than FL118 treatment (Figure 4). On the other hand, in Ba/F3 cells expressing BCR-ABL or its T315I mutant, the treatment with CPT did not suppress the expression of Birc5 mRNA or BIRC5 protein, and the treatment with FL118 only significantly suppressed these expressions (Figure 8). It has been reported that CPT activates the tumor suppressor p53 and that the inactivation of p53 increases the cytotoxicity of CPT in cancer cells [63, 64]. Although p53 in Ba/F3 cells is intact, p53 in K562 cells has a genetic mutation and is inactivated [65]. Therefore, it is possible that CPT had a strong suppressive effect on BIRC5 expression in K562 cells. However, on the other hand, it was previously reported that p53 transcriptionally repress BIRC5 expression by interfering with E2F1 on the BIRC5 promoter or by the recruitment of a corepressor factor Sin-3 and Histone Deacetylases HDAC on the BIRC5 promoter [66, 67]. Now, we have no suitable explanation for this discrepant result. “
<References>
- Gupta, M.; Fan, S.; Zhan, Q.; Kohn, K.W.; O'Connor, P.M.; Pommier, Inactivation of p53 increases the cytotoxicity of camptothecin in human colon HCT116 and breast MCF-7 cancer cells. Clin Cancer Res. 1997, 3, 1653-1660.
- Rudolf, E.; Rudolf, K.; Cervinka, Camptothecin induces p53-dependent and -independent apoptogenic signaling in melanoma cells. Apoptosis. 2011, 16, 1165-1176.
- Law, J.C.; Ritke, M.K.; Yalowich, J.C.; Leder, G.H.; Ferrell, R. Mutational inactivation of the p53 gene in the human erythroid leukemic K562 cell line. Leuk Res. 1993, 17, 1045-1050.
- Hoffman, W.H.; Biade, S.; Zilfou, J.T.; Chen, J.; Murphy, Transcriptional repression of the anti-apoptotic survivin gene by wild type p53. J Biol Chem. 2002, 277, 3247-3257.
- Fischer, M.; Quaas, M.; Nickel, A.; Engeland, Indirect p53-dependent transcriptional repression of Survivin, CDC25C, and PLK1 genes requires the cyclin-dependent kinase inhibitor p21/CDKN1A and CDE/CHR promoter sites binding the DREAM complex. Oncotarget. 2015, 6, 41402-41417.
- Any mechanistic insights on how DDX5 is acting downstream of BCR-ABL? Through transcriptional regulation, effects on RNA processing, etc? Please discuss current knowledge and suggestions for future investigation.
Answer:
We added the following sentences in Discussion in the revised manuscript (p. 19).
“DDX5 has been reported to be involved in RNA metabolisms and to function as a coactivator for certain transcription factors [25-30], and is thought to be deeply related to gene expression regulation at downstream of BCR-ABL in K562 cells. In the future, it will be necessary to analyze how the knockout of DDX5 in K562 cells affects gene expression by RNA sequence analysis and to understand the DDX5-mediated carcinogenesis mechanism activated by BCR-ABL.”
- Is FL118 binding and degrading DDX5 in this CML system? Figure 3 shows it reduces DDX5 protein but a more explicit confirmation would be helpful.
Answer:
We would like to thank the reviewers for their valuable opinions. We received similar comments from another reviewer. In this revised manuscript, we investigated the effect of the proteasome inhibitor MG132 on the FL118-induced decrease in DDX5 expression, and added the results in Figure 3D and the following sentences in the revised manuscript. (p.3-4, p. 18)
“According to previous report by Ling and colleagues, DDX5 was identified as FL118 binding protein, and FL118 behaves as a molecular glue degrader against DDX5 [70]. This report suggests us the FL118-caused downregulation of DDX5 most unlikely due to deceleration of protein synthesis of DDX5. In fact, we observed that the FL118-induced decrease in DDX5 expression was significantly restored by MG132 treatment in K562 cells (Figure 3D). Therefore, FL118 may promote proteasomal degradation of DDX5 in K562 cells. The detailed mechanism how DDX5 is downregulated by the treatment with FL118 will be a problem which should be clarified in next future project.”
<Reference>
- Ling, X.; Wu, W.; Aljahdali, I.A.M.; Liao, J.; Santha, S.; Fountzilas, C.; Boland, P.M.; Li, FL118, acting as a 'molecular glue degrader', binds to dephosphorylates and degrades the oncoprotein DDX5 (p68) to control c-Myc, survivin and mutant Kras against colorectal and pancreatic cancer with high efficacy. Clin Transl Med. 2022, 12, e881.
Minor comments:
- Abbreviate chronic myeloid leukemia as CML after first use.
Answer:
We have corrected the notation.
- Standardize capitalization and italicization of gene/protein names throughout manuscript.
Answer:
We corrected capitalization and italicization of gene/protein names in the revised manuscript.
Round 2
Reviewer 1 Report
Comments and Suggestions for Authors
The manuscript by Takeda et al. provides strong evidence supporting the notion that DDX5 is a crucial therapeutic target in CML. Additionally, it highlights the potential of FL118 as a promising candidate compound for treating BCR-ABL inhibitor-resistant CML.
The authors have addressed all the previous comments. Thus, the manuscript can be accepted in its present form.
Author Response
The manuscript by Takeda et al. provides strong evidence supporting the notion that DDX5 is a crucial therapeutic target in CML. Additionally, it highlights the potential of FL118 as a promising candidate compound for treating BCR-ABL inhibitor-resistant CML.
The authors have addressed all the previous comments. Thus, the manuscript can be accepted in its present form.
Answer: Thank you for the reviewer's valuable feedback.
Reviewer 2 Report
Comments and Suggestions for Authors
Thanks for providing the revised version and addressing most of reviewer comments. Few suggestions:
1) please check revised title. Potential candidate and therapeutic agents are redundant words.
2) other BCR-ABL cell lines suggested earlier are all not CML lines. So, please revise the newly added sentence to ‘leukemic cell lines’.
Best Wishes,
Comments on the Quality of English LanguageAcceptable with minor edits.
Author Response
Thanks for providing the revised version and addressing most of reviewer comments. Few suggestions:
1) please check revised title. Potential candidate and therapeutic agents are redundant words.
Answer: We amended the title into “ FL118 is a potent therapeutic agent against chronic myeloid leukemia resistant to BCR-ABL inhibitors through targeting RNA helicase DDX5”.
2) other BCR-ABL cell lines suggested earlier are all not CML lines. So, please revise the newly added sentence to ‘leukemic cell lines’.
Answer: We would like to thank for the reviewer’s valuable comment. We added sentence to “leukemic cell lines”.